# Adrenergic activation modulates the signal from the Reissner fiber to cerebrospinal fluid-contacting neurons during development

Yasmine Cantaut-Belarif[1], Adeline Orts Del'Immagine[1], Margot Penru[1], Guillaume Pézeron[2], Claire Wyart[1]*, Pierre-Luc Bardet[1]*

[1]Paris Brain Institute, ICM, Inserm U 1127, CNRS UMR 7225, Sorbonne Université, Paris, France; [2]Molecular Physiology and Adaptation (PhyMA - UMR 7221), Muséum National d'Histoire Naturelle, CNRS, Paris, France

**Abstract** The cerebrospinal fluid (CSF) contains an extracellular thread conserved in vertebrates, the Reissner fiber, which controls body axis morphogenesis in the zebrafish embryo. Yet, the signaling cascade originating from this fiber to ensure body axis straightening is not understood. Here, we explore the functional link between the Reissner fiber and undifferentiated spinal neurons contacting the CSF (CSF-cNs). First, we show that the Reissner fiber is required in vivo for the expression of urp2, a neuropeptide expressed in CSF-cNs. We show that the Reissner fiber is also required for embryonic calcium transients in these spinal neurons. Finally, we study how local adrenergic activation can substitute for the Reissner fiber-signaling pathway to CSF-cNs and rescue body axis morphogenesis. Our results show that the Reissner fiber acts on CSF-cNs and thereby contributes to establish body axis morphogenesis, and suggest it does so by controlling the availability of a chemical signal in the CSF.

*For correspondence:
claire.wyart@icm-institute.org
(CW);
pierreluc.bardet@icm-institute.org
(P-LB)

## Introduction

One of the major questions in the study of multicellular organism development is to understand how precise morphogenesis is ensured during embryonic and postembryonic stages while the animal grows into an adult. In particular, this process requires coordination between cell specification signals and the control of the tissue shape (*Chan et al., 2017*). It has recently emerged that the cerebrospinal fluid (CSF) contains many signals important for cell differentiation, and is an important route for the control of morphogenesis (*Fame and Lehtinen, 2020*). The CSF is a complex liquid filling the central nervous system cavities containing a set of diffusible signaling cues guiding neurogenesis (*Lehtinen et al., 2011*) and brain morphology in a tissue autonomous manner (*Kaiser et al., 2019*; *Langford et al., 2020*). The CSF circulation is in part generated by the coordinated movement of cilia projecting to the lumen of brain ventricles and central canal of the spinal cord (*Faubel et al., 2016*; *Hagenlocher et al., 2013*; *Olstad et al., 2019*; *Sternberg et al., 2018*; *Thouvenin et al., 2020*).

The role of cilia in the control of CSF composition and circulation has recently gained a special attention in zebrafish. Indeed, disruption of cilia motility in this species has been long known to induce a typical phenotype consisting in a downward curvature of the posterior axis of the embryo (*Brand et al., 1996*; *Jaffe et al., 2016*; *Kramer-Zucker et al., 2005*; *Sullivan-Brown et al., 2008*). This phenotype has recently been linked to a role of the CSF in body axis curvature establishment and maintenance downstream of cilia function (*Cantaut-Belarif et al., 2018*; *Grimes et al., 2016*). We previously showed that the Reissner fiber (RF), a component of the CSF, is important for axis

morphogenesis in the zebrafish embryo (*Cantaut-Belarif et al., 2018*). The Reissner fiber (Reissner, 1860) is an acellular proteinous thread bathing in the brain and spinal cord cavities early in development (*Troutwine et al., 2020*), formed by the aggregation of the SCO-spondin glycoprotein, secreted into the CSF by the subcommissural organ and the floor plate (*Lehmann and Naumann, 2005*; *Meiniel et al., 2008*). This fiber fails to assemble in cilia-defective mutants, and *scospondin* mutant embryos lacking the Reissner fiber exhibit a curled-down axis despite exhibiting normal cilia motility and CSF flow (*Cantaut-Belarif et al., 2018*; *Rose et al., 2020*). These observations suggest that the fiber itself plays a role in the control of body axis morphogenesis in the embryo, and that its absence causes a curled-down phenotype, including in cilia-defective mutants. However, the signal linking the presence of the Reissner fiber and its long-range effect on body axis morphogenesis is not fully understood.

Recent studies suggest that bath applications of monoamines can restore a straight axis in embryos with a curled-down phenotype (*Lu et al., 2020*; *Zhang et al., 2018*). Monoamines are widespread neurotransmitters and neuromodulators that include epinephrine and norepinephrine, and influence many neurophysiological processes in the adult such as sleep control (*Singh et al., 2015*). Radiolabeled norepinephrine can bind the Reissner fiber in the CSF of rats (*Caprile et al., 2003*) and frogs (*Diederen et al., 1983*). The study of their role in development is emerging, with a special focus on neuronal precursors proliferation and differentiation (*Berg et al., 2013*).

In addition, the signal controlling axis morphogenesis downstream of CSF flow has been associated with changes in the expression of Urp1 and Urp2 peptides (*Lu et al., 2020*; *Zhang et al., 2018*). These peptides belong to a family of neuropeptides similar to Urotensin-II previously shown to mediate endocrine, cardiac, and neurophysiological functions (*Vaudry et al., 2015*). In the zebrafish embryo, *urp1* and *urp2* are expressed along the antero-posterior axis of the neural tube in a subset of spinal sensory neurons (*Quan et al., 2015*) called CSF-contacting neurons (CSF-cNs). CSF-cNs extend into the CSF a microvilliated apical extension that starts differentiating in the embryo and becomes fully mature in the larva to tune the mechanosensory responses of CSF-cNs upon tail bending (*Desban et al., 2019*). These conserved interoceptive neurons form two populations of different developmental origins that are located ventrally and dorso-laterally relative to the central canal of the zebrafish spinal cord (*Djenoune et al., 2017*; *Park et al., 2004*). At the embryonic stage, undifferentiated CSF-cNs that are located ventrally in the neural tube are the only ones expressing Urp neuropeptides and exhibiting spontaneous calcium transients at rest (*Sternberg et al., 2018*). Recently, the Reissner fiber was shown to be required in the larva for the mechanosensory function leading to intracellular calcium increase in differentiated CSF-cNs (*Orts-Del'Immagine et al., 2020*). However, the role of the Reissner fiber in CSF-cN spontaneous activity at the embryonic stage when they express Urp neuropeptides is unknown.

In this study, we aimed to better understand the signal linking the Reissner fiber to body axis morphogenesis. To investigate whether the Reissner fiber is required for a signal toward the immature CSF-cNs, we sought to decipher the nature of this signal and how it affects at long distance body axis straightening. We show that the Reissner fiber is required for a signal controlling *urp2* expression. Using in vivo calcium imaging, we report that the Reissner fiber is also required for calcium signaling in *urp2*-expressing ventral CSF-cNs, confirming the existence of a crosstalk between the Reissner fiber and undifferentiated CSF-cNs at the embryonic stage. Using the *pkd2l1* mutant, we show that the loss of calcium signaling in ventral CSF-cNs does not lead to a loss of *urp2* expression nor embryonic body axis curvature. Finally, we show that epinephrine and norepinephrine can restore the Reissner fiber-dependent signal when injected locally in the brain ventricles, and can restore body axis defects of *scospondin* mutants. Our work demonstrates that the Reissner fiber-dependent signal to ventral CSF-cNs contributes to body axis straightening and is modulated by adrenergic ligands.

## Results

### The Reissner fiber controls *urp2* gene expression

To explore the signals downstream the Reissner fiber, we performed a transcriptomic analysis of *scospondin* mutants (*Figure 1*).

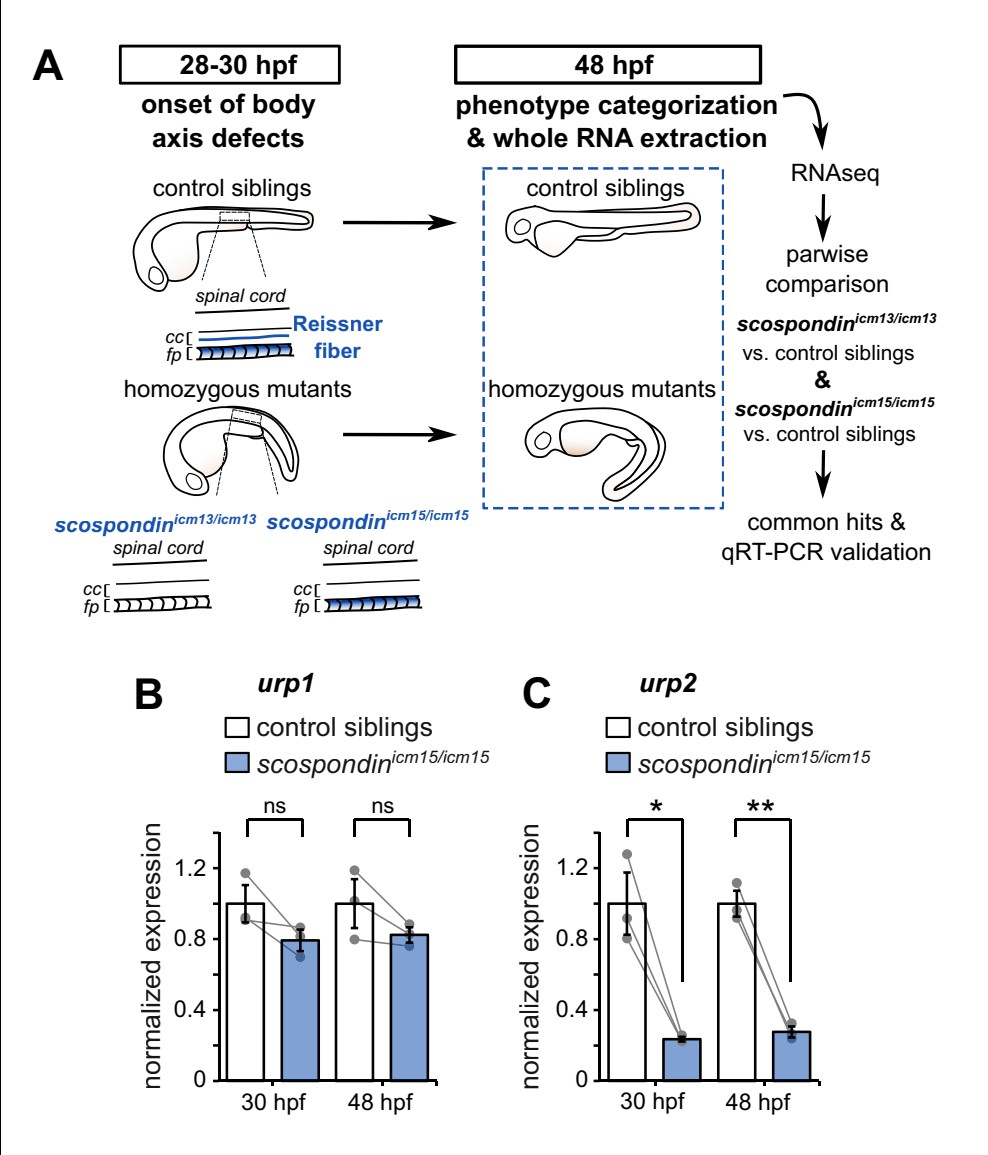

**Figure 1.** The Reissner fiber is required for *urp2* but not *urp1* gene expression. (**A**) Schematic of the experimental design. Embryos obtained from *scospondin*^icm15/+ or *scospondin*^icm13/+ incrosses were raised until 48 hpf and categorized according to their external phenotype: straight body axis (control siblings, top) or curled-down body axis (homozygous mutants, bottom) prior to RNA extraction. RNA sequencing was performed on three independent replicates for each allele and allowed pairwise comparisons of transcriptomes to identify commonly regulated genes. qRT-PCR experiments were performed at 30 hpf to validate transcriptomic data at the onset of body axis defects induced by the loss of the Reissner fiber and at 48 hpf when the phenotype is fully developed (48 hpf). Null *scospondin*^icm13/icm13 and hypomorphic *scospondin*^icm15/icm15 mutant embryos share the same peculiar curled-down phenotype induced by the loss of the Reissner fiber in the central canal of spinal cord (cc). However, *scospondin*^icm15/icm15 mutants retain SCO-spondin protein expression in secretory structures such as the floor plate (fp). (**B, C**) qRT-PCR analysis of mRNA levels of *urp1* (**B**) and *urp2* (**C**) in *scospondin*^icm15/icm15 mutants (blue) compared to their control siblings (white) at 30 and 48 hpf. Data are represented as mean ± SEM. N = 3 independent biological replicates for each condition. Each point represents a single experimental replicate. ns p>0.05, *p<0.05, **p<0.01 (paired t-test). See also *Figure 1—figure supplement 1* and *Figure 1—source data 1* and *2*.

The online version of this article includes the following source data and figure supplement(s) for figure 1:

**Source data 1.** Data for *Figure 1B–C*.
**Source data 2.** Up- and down-regulated transcripts in curled-down *scospondin* mutants compared to their control siblings.
*Figure 1 continued on next page*

*Figure 1 continued*

**Figure supplement 1.** Time course of the evolution of body axis geometry from 20 to 30 hpf in *scospondin*$^{icm15/icm15}$ mutants.

We took advantage of two previously-generated *scospondin* alleles (*Cantaut-Belarif et al., 2018*) to evaluate transcriptional modifications associated with the curled-down phenotype due to the loss of the Reissner fiber in the CSF. While the *scospondin*$^{icm13}$ null allele leads to a dual loss of the fiber in the central canal and of the SCO-spondin protein detection in secretory structures, the *scospondin*$^{icm15}$ hypomorphic allele retains protein expression but solely precludes the Reissner fiber formation (*Figure 1A*). Both homozygous mutants lack the Reissner fiber and exhibit a typical curled-down phenotype, which arises from 28 to 30 hr post-fertilization (hpf) onwards and was undetectable beforehand (*Cantaut-Belarif et al., 2018*, *Figure 1A*, *Figure 1—figure supplement 1*). We performed pairwise comparisons of the transcriptomes of homozygous mutants *versus* control siblings at 48 hpf when the body axis curvature defect is fully penetrant (*Cantaut-Belarif et al., 2018*). The resulting lists of up- and down-regulated transcripts that were common to the two *scospondin* alleles are presented in *Figure 1—source data 2*. Very few genes exhibited an important change in expression, and only a handful changed more than two folds. Noticeably, we observed a strong reduction of *urp2* gene expression in curled-down mutant embryos lacking the Reissner fiber compared to their control siblings (mean ± SEM fold decrease: 4.07 ± 1.2 in *scospondin*$^{icm13/icm13}$ and 4.69 ± 1.77 in *scospondin*$^{icm15/icm15}$, n = 3 replicates each, p-value<0.00005, GLM test; see Materials and methods and *Figure 1—source data 2* for details). There was no significant decrease in *urp1* transcript levels (mean ± SEM fold decrease: 0.97 ± 0.06 in *scospondin*$^{icm13/icm13}$ and 1.22 ± 0.25 in *scospondin*$^{icm15/icm15}$, n = 3 replicates each, p-value=p = 0.191, GLM test).

The gene *urp2* encodes for a secreted neuropeptide belonging to the Urotensin-II-related-peptide family (*Tostivint et al., 2014*). Together with *urp1*, these transcripts have recently been identified as both strongly downregulated in curled-down mutants (*Lu et al., 2020*; *Zhang et al., 2018*). To ascertain our RNAseq results and confirm the difference we observed with previous results, we carried out qRT-PCR analysis of *urp1* and *urp2* expression levels in the hypomorphic *scospondin*$^{icm15}$ allele. Interestingly, we observed that *urp1* expression level is not significantly decreased in *scospondin* homozygous mutants compared to their control siblings, neither at 30 hpf nor at 48 hpf (*Figure 1B*). Consistently with transcriptomic results, *urp2* expression level shows a strong decrease at 48 hpf in mutant embryos compared to their control siblings (3.6 ± 0.2 fold decrease; mean ± SEM; *Figure 1C*). This is also true at the onset of the curled-down phenotype (30 hpf: 4.2 ± 0.5 fold decrease; mean ± SEM; *Figure 1C*) indicating that *urp2* gene expression level is affected when embryos start to develop an abnormal morphogenesis of the posterior axis. Taken together, these data show that in zebrafish embryo the presence of the Reissner fiber in the CSF is required for the normal expression level of *urp2,* but not *urp1*.

## The Reissner fiber is required for calcium signaling in *urp2* expressing CSF-cNs

The expression of Urotensin-II-related peptides is restricted to the ventral population of CSF-cNs (*Quan et al., 2015*), known to exhibit spontaneous intracellular calcium variations around 30 hpf (*Sternberg et al., 2018*) when the curled-down phenotype becomes visible in *scospondin* mutants (*Cantaut-Belarif et al., 2018* and *Figure 1—figure supplement 1*). Curled-down cilia-defective embryos lack these early calcium transients (*Sternberg et al., 2018*) and do not form a proper Reissner fiber (*Cantaut-Belarif et al., 2018*). We therefore hypothesized that the Reissner fiber may functionally interact with ventral CSF-cNs that are expressing *urp2*.

To address this question, we performed in vivo population calcium imaging at 28–30 hpf using the *Tg(pkd2l1:GCaMP5G)* line labeling the dorso-lateral and ventral CSF-cNs in the spinal cord (*Figure 2A*, *Figure 2—video 1*). As previously described, we observed that ventral CSF-cNs exhibit spontaneous calcium transients in wild-type embryos (*scospondin*$^{+/+}$, *Figure 2B*, *Figure 2—video 1*). Quantification of the integrated fluorescence variations over time showed that in heterozygous *scospondin*$^{icm15/+}$ embryos, which display a straight body axis and form a proper Reissner fiber (*Cantaut-Belarif et al., 2018*), ventral CSF-cNs retained the same level of activity than in wild-type

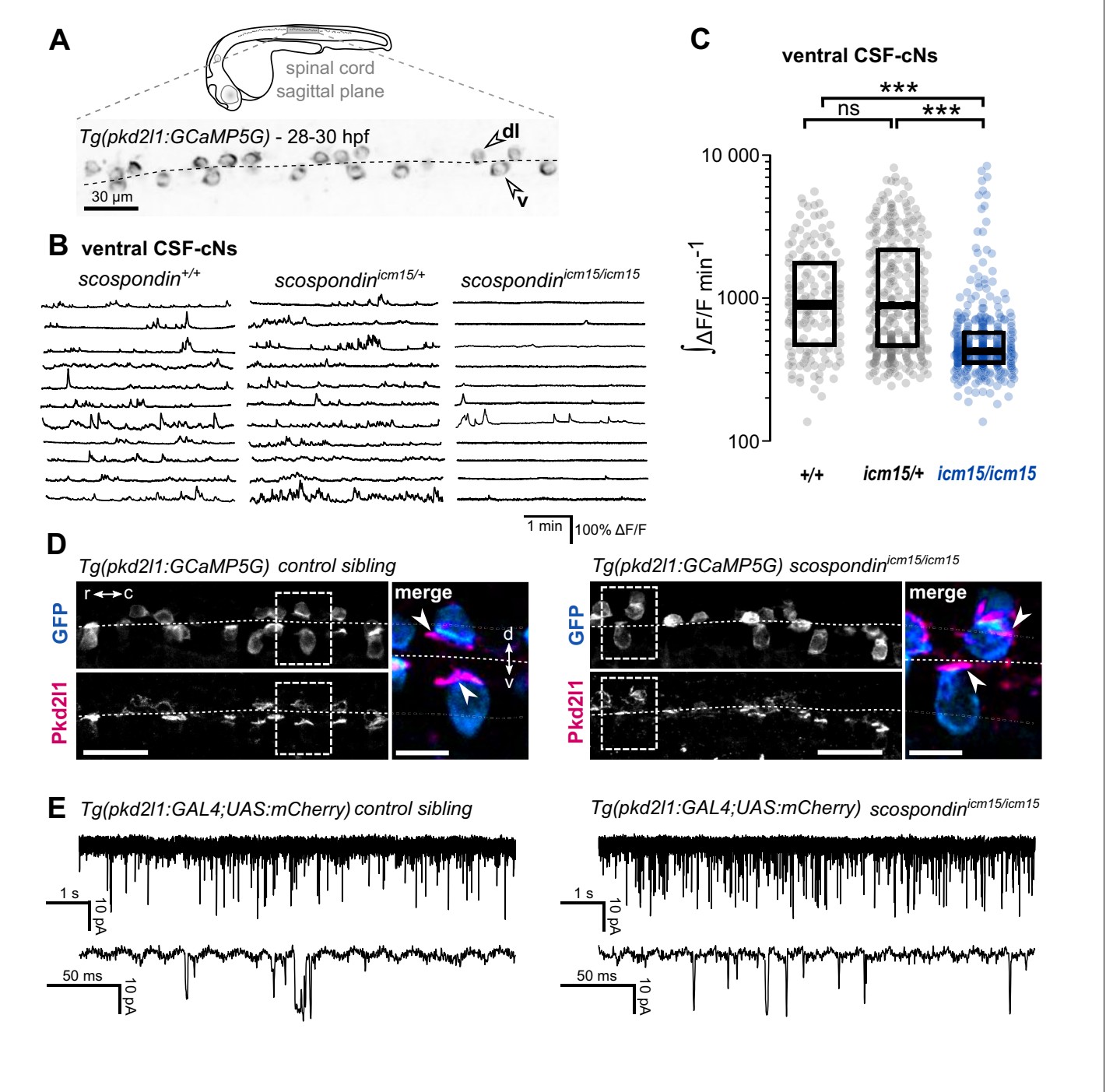

**Figure 2.** The Reissner fiber is required for the spontaneous calcium variations of ventral CSF-contacting neurons. (A) 28–30 hpf embryos expressing the GCaMP5G calcium reporter in CSF-contacting neurons were imaged on the lateral side. *Tg(pkd2l1:GCaMP5G)* embryos label both dorso-lateral (dl, above the dotted line) and ventral (v, below the dotted line) CSF-contacting neurons (arrowheads). Scale bar: 30 µm. (B) Representative traces of calcium variations in individual ventral CSF-contacting neurons in wild-type (*scospondin*^+/+^), heterozygous (*scospondin*^icm15/+^) and *scospondin*^icm15/icm15^ mutants. Sample traces from individual cells with integral ΔF/F values ranging around the median distribution of the imaged population are represented for each genotype (n = 11 cells). (C) Quantification of the normalized integrated calcium variation over time of ventral CSF-contacting neurons in wild-type (+/+), heterozygous (*icm15/+*) and *scospondin*^icm15/icm15^ mutants (*icm15/icm15*, blue). Data were collected from five independent experiments and include 10 wild-type embryos (n = 146 cells), 20 heterozygous embryos (n = 287 cells) and 21 *scospondin*^icm15/icm15^ mutants (n = 307 cells). Each point represents a single cell. Bottom and top edges of the boxes indicate the 1st and 3rd quartiles. Bold lines represent the median value for each distribution. ns p>0.05, ***p<0.001 (Kolmogorov-Smirnov test). (D) Immunohistochemistry for Pkd2l1 (magenta) and GFP (blue) in *Tg(pkd2l1: GCaMP5G)* embryos at 30 hpf in the spinal cord of a control sibling (left) and *scospondin*^icm15/icm15^ mutant (right). Scale bar: 30 µm. Magnification of the

*Figure 2 continued on next page*

*Figure 2 continued*

area delineated by dotted line boxes is represented for each condition (r: rostral, c: caudal, d: dorsal, v: ventral). Scale bar: 10 µm. *scospondin*[icm15/icm15] embryos show a similar localization of the Pkd2l1 protein at the developing apical extension (arrowheads) of the CSF-cNs (labeled by the GFP antibody, blue) compared to control siblings. (E) In vivo voltage-clamp recordings from CSF-contacting neurons in the *Tg(pkd2l1:GAL4;UAS:mCherry)* line at 30 hpf in control embryos (left) and *scospondin*[icm15/icm15] mutants (right). Note the extensive number of events in both conditions (top traces). Bottom traces represent higher temporal magnifications and allow distinguishing single channel openings. See also *Figure 2—figure supplement 1* and *Figure 2—video 1*.

The online version of this article includes the following video, source data, and figure supplement(s) for figure 2:

**Source data 1.** Data for *Figure 2C* and *Figure 2—figure supplement 1B–C*.
**Figure supplement 1.** The spontaneous activity of dorso-lateral CSF-contacting neurons does not require the Reissner fiber in the embryo.
**Figure 2—video 1.** Intracellular calcium transients of ventral CSF-contacting neurons are reduced in *scospondin* mutants.
https://elifesciences.org/articles/59469#fig2video1

(*Figure 2C*). On the contrary, homozygous *scospondin*[icm15/icm15] embryos exhibited a 52.8% decrease of calcium activity compared to wild-type (*Figure 2C*). A similar reduction of 39% occurred in the null *scospondin*[icm13/icm13] mutant compared to wild-type siblings (*Figure 2—figure supplement 1A,B*). These data show that the loss of the Reissner fiber reduces spontaneous calcium variations of ventral CSF-cNs during the critical time window for body axis straightening. On the contrary, the sparse spontaneous calcium transients in dorsal CSF-cNs were not affected by the absence of the Reissner fiber in both *scospondin* alleles (*Figure 2—figure supplement 1C,D*).

Early calcium transients in CSF-cNs require the non-selective cationic channels permeable Pkd2l1 channel in vivo (*Sternberg et al., 2018*) that is permeable to calcium ions. The reduction in calcium activity when the Reissner fiber is lacking could therefore be due to a defect in the Pkd2l1 channel localization or opening probability. *In toto* immunohistochemistry for Pkd2l1 protein showed that this channel is enriched at the differentiating apical extension of CSF-cNs in curled-down *scospondin*[icm15/icm15] mutants as well as in control embryos (*Figure 2D*). In order to assess Pkd2l1 channel properties, we performed in vivo whole-cell voltage-clamp recordings in double transgenic *Tg(pkd2l1:GAL4;UAS:mCherry)* embryos. In the absence of a Reissner fiber in *scospondin*[icm15/icm15] mutants, we observed spontaneous Pkd2l1 channel openings that were similar to control embryos (*Figure 2E*). Thus, the loss of the Reissner fiber decreases CSF-cNs intracellular calcium variations without preventing the opening nor the localization of Pkd2l1 channels in the differentiating apical extension of the cells. Altogether, these data suggest that the Reissner fiber is required for a signal acting on ventral CSF-cNs that controls both the spontaneous calcium variations and the expression of *urp2* in the embryo.

## Loss of *urp2* expression is only observed in mutants devoid of the Reissner fiber, independently of calcium variations in CSF-cNs

As the Reissner fiber is required for both ventral CSF-cNs spontaneous calcium transients and *urp2* gene expression, we tested whether intracellular calcium variations are necessary for a normal *urp2* expression and axis straightness. We took advantage of the *pkd2l1*[icm02/icm02] mutant where the embryonic activity of CSF-cNs is abolished (*Sternberg et al., 2018*).

Thanks to the viability of *pkd2l1*[icm02/icm02] zygotic mutants, we generated genetically-related clutches that were either fully wild-type or fully maternal and zygotic (MZ) homozygous mutant (*Figure 3A*). We observed that *pkd2l1*[icm02/icm02] MZ mutant embryos did not display any defect in body axis morphogenesis and were morphologically indistinguishable from wild-type embryos (*Figure 3B*). Using immunohistochemistry, we observed that *pkd2l1*[icm02/icm02] mutants form a normal Reissner fiber in the central canal of the spinal cord at 30 hpf (*Figure 3C*). Next, using qRT-PCR, we tested whether *urp1* and *urp2* gene expression levels were diminished by the absence of calcium transients in CSF-cNs (*Figure 3D*). As the mutation in the *icm02* allele generates a premature stop codon in the *pkd2l1* gene (*Böhm et al., 2016*), one can predict that the *pkd2l1* mRNA would be degraded by nonsense-mediated decay. Indeed, mutant clutches displayed a 4.9 fold decrease in *pkd2l1* transcripts level compared to wild-type counterparts (*Figure 3D*). Interestingly, we observed that *urp2* and *urp1* gene expression were not decreased in *pkd2l1*[icm02/icm02] embryos compared to wild-types (*Figure 3D*). Altogether, these data show that the loss of Pkd2l1-driven calcium transients in CSF-cNs does not lead to a decreased expression of *urp1* and *urp2* transcripts, rejecting the

hypothesis of a direct requirement for *urp2* expression of Pkd2l1-dependent intracellular calcium variations. Instead, our results show a strict correlation between the presence of the Reissner fiber, a normal *urp2* expression level and the proper morphogenesis of the embryonic body axis. Altogether, our results prompted us to hypothesize that the aberrant posterior axis curvature in absence of the Reissner fiber is a consequence of the decreased *urp2* expression.

### *urp2* expression in the absence of the Reissner Fiber can restore posterior axis defects

Given the strict correlation between *urp2* expression and body axis straightening, we hypothesized that restoring *urp2* levels would rescue the posterior axis curvature developed by *scospondin*^icm15/icm15^ mutants. To test this hypothesis, we performed one cell stage *urp2* mRNA injections on clutches obtained from *scospondin*^icm15/+^ parents. We sorted the injected embryos at 48 hpf into two morphological categories: curled-down and non-curled-down (*Figure 3E and F*). In the four independent experiments conducted, injections with a control mRNA led to a proportion of curled-down embryos close to 25%, as expected (*Figure 3F*). On the contrary, embryos overexpressing *urp2* showed much lower proportions of curled-down embryos (10.7 ± 1.5%; mean ± SEM; *Figure 3F*), suggesting that some homozygous mutants are rescued in this condition. To confirm this rescue, we genotyped curled-down and non-curled-down 48 hpf embryos injected with a control mRNA or *urp2* mRNA (*Figure 3E and G*, *Figure 3—figure supplement 1*). As expected, wild-type and heterozygous embryos injected with the control mRNA displayed a straight body axis, while *scospondin*^icm15/icm15^ showed a downward curvature of the posterior axis. Instead, *urp2* overexpression lead to detect straight or slightly curled-up *scospondin*^icm15/icm15^ embryos at several instances (*Figure 3E and G*, *Figure 3—figure supplement 1*). Thus, restoring higher *urp2* levels is sufficient to prevent the embryonic posterior axis defects in *scospondin* mutants. These observations confirm that *urp2* neuropeptide expressed in CSF-cNs may signal at long-range, as suggested from the expression of the receptor in dorsal somites (*Zhang et al., 2018*), to ensure a proper axis morphogenesis.

### Epinephrine and norepinephrine restore morphogenesis of the posterior axis and *urp* expression in *scospondin* mutants

Based on our observations, we assumed that the Reissner fiber is necessary for the activity of at least one signaling pathway regulating *urp2* gene expression together with body axis morphogenesis. Epinephrine and norepinephrine belong to the monoamine neurotransmitter family and are known to bind the Reissner fiber in rats (*Caprile et al., 2003*) and frogs (*Diederen et al., 1983*). Recently, systemic bath applications of monoamines have been described to rescue body axis defects in curled-down cilia-defective mutants (*Lu et al., 2020*), (*Zhang et al., 2018*). We investigated the possible role of epinephrine and norepinephrine in the regulation of the RF signaling pathways.

First, we tested if epinephrine and norepinephrine could influence the curled-down phenotype developed by *scospondin*^icm15/icm15^ mutants. We compared sibling animals generated from incrosses of *scospondin*^icm15/+^. We first analyzed the effect of 2.5 hr bath applications of epinephrine and norepinephrine on control embryos at 30 hpf (*Figure 4A*). Epinephrine and norepinephrine have a moderate impact on the shape of the head-to-tail axis of initially straight embryos, which display a slight curled-up phenotype after exposure to monoamines (*Figure 4B*). To estimate the straightness of the posterior axis, we quantified the angle formed between the ear, the caudal limit of the yolk extension and the tip of the tail (*Figure 4C*, top panel). This angle is distributed around 190.6° in control embryos exposed to a vehicle solution (median value; *Figure 4C*, bottom left graph). Comparatively, control embryos exposed to monoamines exhibit a distribution of ear-to-tail angles shifted toward slightly higher values (median values: 220.4° and 213.6° for epinephrine and norepinephrine, respectively; *Figure 4C*, bottom right graph). Next, we analyzed the effect of epinephrine and norepinephrine on *scospondin*^icm15/icm15^ embryos. While mutant animals exposed to a vehicle solution display a typical curled-down body axis, *scospondin*^icm15/icm15^ mutants treated with monoamines exhibit a reduction in the downward curvature of the posterior axis (*Figure 4B*). Quantifications of ear-to-tail angles in *scospondin*^icm15/icm15^ mutants show a large increase of the median angle value after exposure to monoamines, compared to embryos treated with a vehicle solution (106.5° for vehicle, 167.1° for epinephrine and 158.5° for norepinephrine; median values; *Figure 4C*). Altogether, these data

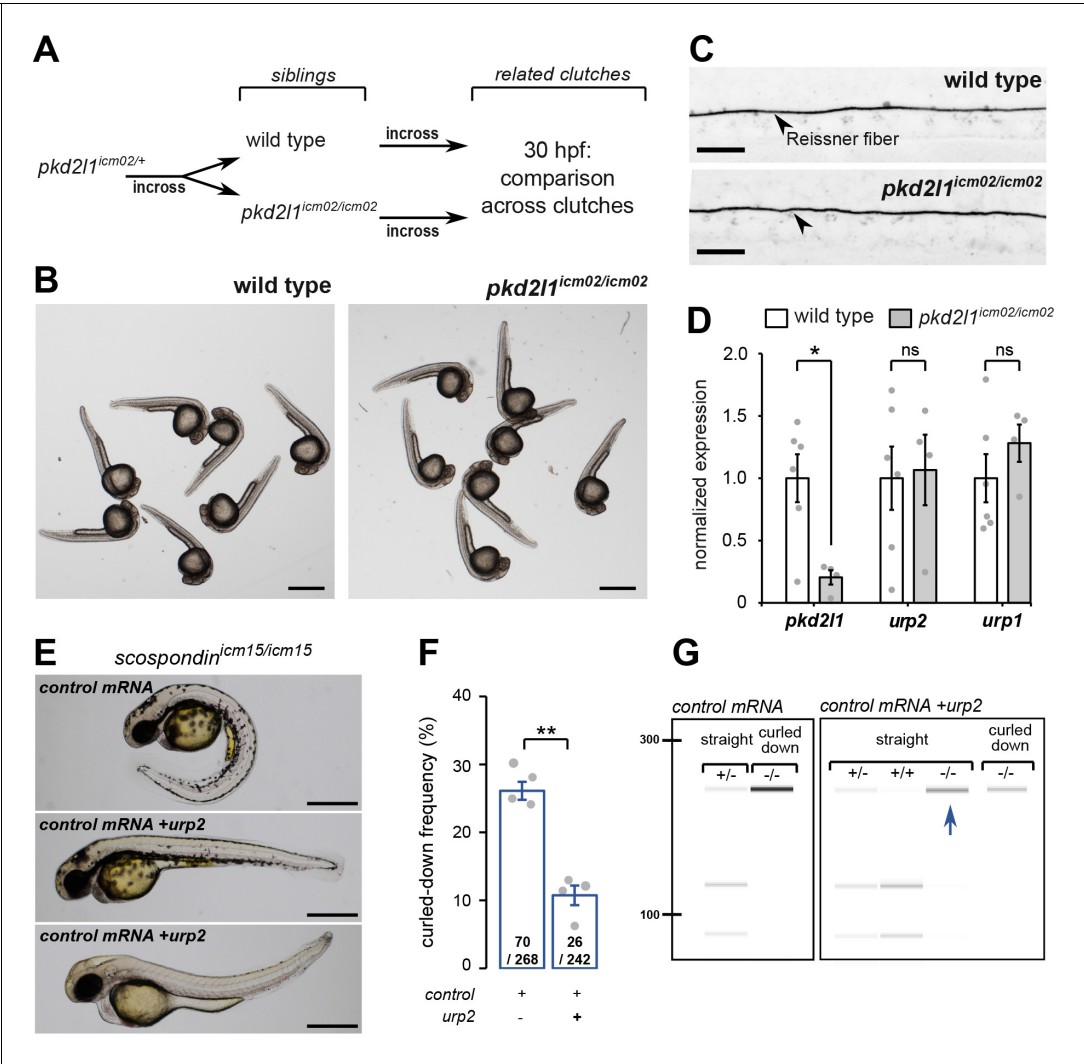

**Figure 3.** *urp2* expression is Pkd2l1-independent and is important for the Reissner fiber-dependent straightening of the embryonic posterior axis. (A) Adult wild-type and *pkd2l1*[icm02/icm02] siblings were incrossed to generate related clutches that were analyzed at 30 hpf. (B) Representative pictures of wild-type (left) and *pkd2l1*[icm02/icm02] embryos (right) at 30 hpf. Note that mutant embryos develop a straight posterior axis. Scale bar: 0.5 mm. (C) Representative immunohistochemistry for the Reissner fiber imaged from the spinal cord of a wild-type (top, one representative embryo out of 20) and a *pkd2l1*[icm02/icm02] embryo (bottom, one representative embryo out of 20). Note that the Reissner fiber forms properly in the mutant. Scale bars: 10 μm. (D) qRT-PCR analysis of mRNA levels of *pkd2l1*, *urp2*, and *urp1* in wild-type (white) and *pkd2l1*[icm02/icm02] embryos (grey). Data are represented as mean ± SEM. N = 6 independent replicates for wild-type and four for *pkd2l1*[icm02/icm02]. Each point represents a single experimental replicate. ns p>0.05, *p<0.05 (unpaired t-test). (E) Representative pictures of *scospondin*[icm15/icm15] mutant embryos at 48 hpf after one cell stage injections of a control mRNA alone or of a mix containing a control mRNA and *urp2* mRNA (middle and bottom). Note that upon control injections, *scospondin*[icm15/icm15] mutants display at typical curled-down phenotype, while *urp2* overexpression can lead to straightened (middle) or slightly curled-up posterior axis (bottom). Scale bar: 0.5 mm. (F) Quantification at 48 hpf of curled-down frequency in embryos obtained from *scospondin*[icm15/+] incrosses upon control mRNA injections (n = 70 curled-down animals out of 268) or *urp2* mRNA overexpression (26 curled-down embryos out of 242). Data were collected from four independent clutches and represented as mean ± SEM. **p<0.01 (paired t-test). (G) Injected embryos were genotyped at 48 hpf based on the loss of a restriction site in the *scospondin* mutant allele leading to a band resistant to digestion (-/-). While mutant animals are exclusively curled-down in control conditions, *urp2* mRNA overexpression leads to the detection of mutant animals displaying a straight body axis (blue arrow). See also *Figure 3—figure supplement 1*.

The online version of this article includes the following source data and figure supplement(s) for figure 3:

**Source data 1.** Data for *Figure 3D*.

**Figure supplement 1.** *urp2* overexpression recues body axis curvature defects in *scospondin* mutants.

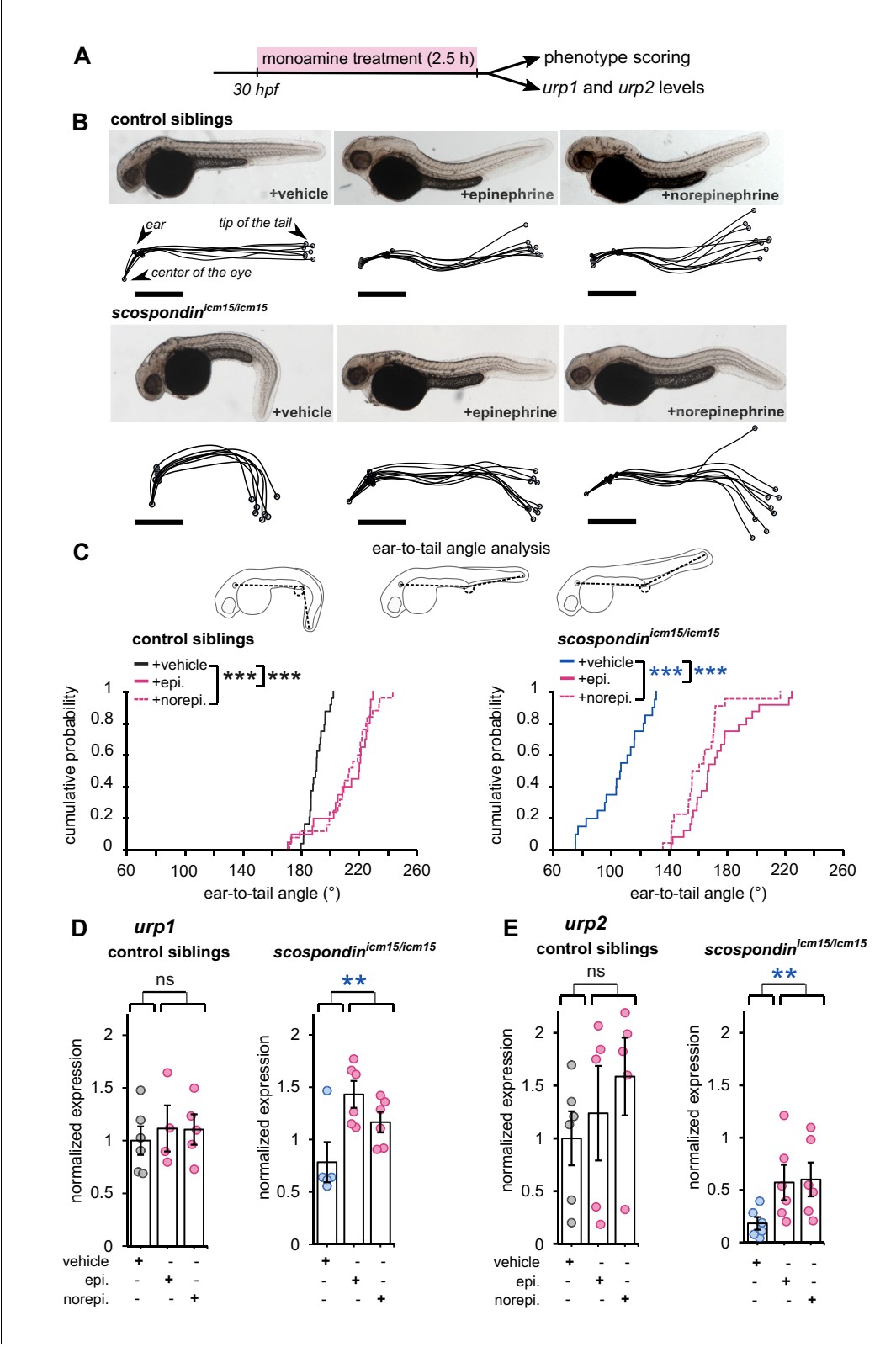

**Figure 4.** Epinephrine and norepinephrine compensate the loss of the Reissner fiber for body axis straightening and increase *urp* expression. (A) Curled-down *scospondin*^icm15/icm15 mutants and their control siblings were sorted at 30 hpf according to the geometry of their posterior axis and then exposed to a E3 solution (vehicle), epinephrine or norepinephrine for 2.5 hr prior to phenotype scoring and RNA extraction. (B) Representative pictures of control siblings (top) and *scospondin*^icm15/icm15 mutants (bottom) after vehicle (left), epinephrine (middle) or norepinephrine (right) treatments. For
*Figure 4 continued on next page*

*Figure 4 continued*

each condition, the global morphologies of treated embryos are represented by superimposed traces linking the center of the eye, the ear and the tip of the tail in one representative clutch. Scale bar: 0.5 mm. (C) Quantification of the angle formed between the ear, the caudal limit of the yolk extension and the tip of the tail (as shown on the schematics, top) in control siblings (bottom left) and *scospondin^{icm15/icm15}* mutants (bottom right). Data were collected from three independent experiments and include 24, 20, 25 control siblings treated with a vehicle solution, epinephrine, norepinephrine respectively (black, solid pink, and dotted pink line respectively) and 20, 24, 22 *scospondin^{icm15/icm15}* embryos treated with a vehicle solution, epinephrine, norepinephrine respectively (blue, solid pink, and dotted pink line respectively) ***p<0.001 (Kolmogorov-Smirnov test). (D, E) qRT-PCR analysis of the mRNA level of *urp1* (D) and *urp2* (E) in control siblings (left) and *scospondin^{icm15/icm15}* embryos (right). Data are represented as mean ± SEM. n = 4 to 6 independent replicates for each condition. Each point represents a single experimental replicate. ns p>0.05, **p<0.01 (GLM test). The online version of this article includes the following source data for figure 4:

**Source data 1.** Data for *Figure 4C–E*.

show that epinephrine and norepinephrine can partially restore the posterior axis geometry of *sco-spondin^{icm15/icm15}* mutants, suggesting that monoamines can rescue the Reissner fiber-dependent signal required for a straight embryonic body axis.

Next, we analyzed the effect of epinephrine and norepinephrine on *urp1* and *urp2* gene expression using qRT-PCR. In control siblings, *urp1* expression remains comparable after vehicle, epinephrine or norepinephrine exposure (*Figure 4D*). Consistently with our previous results, *urp1* expression is not significantly modified in curled-down *scospondin^{icm15/icm15}* embryos compared to their control siblings receiving the same treatment (1.01 ± 0.17-fold change; mean ± SEM; *Figure 4D*). However, we observed a slight increase of *urp1* expression in curled-down mutant embryos treated with epinephrine and norepinephrine compared to vehicle treatment (1.58 ± 0.43 and 1.28 ± 0.23 fold changes for epinephrine and norepinephrine respectively; mean ± SEM). As expected, *urp2* expression level is significantly decreased in curled-down *scospondin^{icm15/icm15}* mutants compared to straight siblings (0.22 ± 0.06-fold change; mean ± SEM; *Figure 4E*). Epinephrine and norepinephrine treatments also do not change *urp2* expression in control siblings, but significantly increase it in curled-down homozygous mutant embryos (7.42 ± 4,23 and 4.67 ± 1.15 fold changes for epinephrine and norepinephrine respectively; mean ± SEM; *Figure 4E*).

These observations show that the rescue of the posterior axis curvature of *scospondin* homozygous mutant embryos by monoamines is associated with an increase of the expression of *urp1* and *urp2* neuropeptides. Epinephrine and norepinephrine compensate the loss of the Reissner fiber both on Urotensin-II-related neuropeptides expression and on posterior axis curvature, suggesting that these compounds act on the Reissner fiber-dependent signaling pathway in the embryonic CSF.

## Epinephrine and norepinephrine restore the Reissner fiber-dependent calcium signaling in ventral CSF-cNs of *scospondin* mutants

The Reissner fiber is required for three concomitant events: intracellular calcium variations and *urp2* expression in ventral CSF-cNs, and the straightening of the posterior axis. We therefore asked whether the delivery of monoamines in CSF could also restore calcium variations in ventral CSF-cNs. To address this question, we performed hindbrain ventricle injections of epinephrine or norepinephrine in the *Tg(pkd2l1:GCaMP5G); scospondin^{icm15}* embryos at 30 hpf, and recorded calcium variations in ventral CSF-cNs within the spinal cord 20 to 60 min after the injections (*Figure 5A*). In order to assess how the monoamines used diffuse in the central canal of the spinal cord and whether they are stable enough in the CSF, we performed immunostainings against norepinephrine 30 and 60 min after hindbrain ventricle injections (*Figure 5—figure supplement 1*). We observed an abundant norepinephrine-positive signal in the central canal of the rostral most region of the spinal cord 30 min post-injection, and the median and caudal most-regions 60 min post-injection, which was absent in control injections (*Figure 5—figure supplement 1*). This observation indicates that norepinephrine diffuses down the central canal when injected in the same conditions used for in vivo calcium imaging and that it is stable in the CSF after an hour, in agreement with former results in the mammalian ventricular system of dogs (*Maas and Landis, 1965*), sheeps (*Forbes and Baile, 1974*) and rats (*Fuxe and Ungerstedt, 1966*; *Levitt et al., 1983*).

We first analyzed the effect of epinephrine and norepinephrine injections on control embryos displaying a straight body axis at 30 hpf and observed that an exogenous delivery of monoamines did not influence the basal calcium variations of ventral CSF-cNs (*Figure 5B and D*). Next, we analyzed

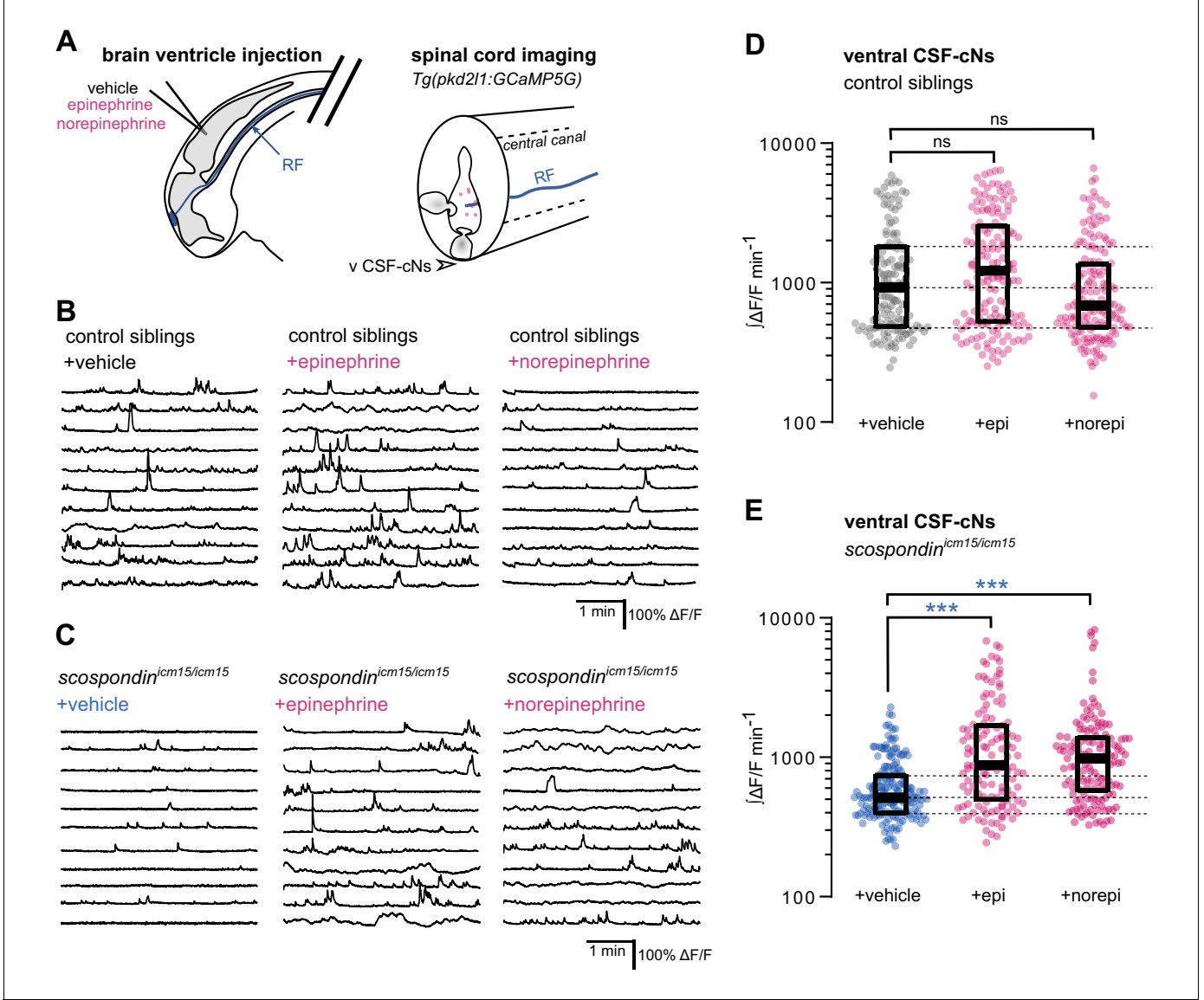

**Figure 5.** Local monoamine delivery restores calcium variations of ventral CSF-contacting neurons in *scospondin* mutants. (A) *Tg(pkd2l1:GCaMP5G)* embryos we used to perform hindbrain ventricle injections at 30 hpf of artificial CSF (vehicle), epinephrine or norepinephrine (left). Intracellular calcium variations in ventral CSF-contacting neurons (v CSF-cNs, arrowhead) were recorded in the spinal cord 30 min after the injection to allow monoamines (pink dots) diffusing down the central can where bathes the Reissner fiber (RF) in control embryos. (B, C) Representative traces of calcium variations of individual ventral CSF-contacting neurons in control siblings (B) and *scospondin*^icm15/icm15^ mutants (C) after vehicle (left), epinephrine (middle) and norepinephrine injections (right). Sample traces from individual cells with integral ΔF/F values ranging around the median distribution of the imaged population are represented for each condition (n = 11). (D, E) Quantification of the normalized integrated intracellular calcium variation over time of ventral CSF-contacting neurons in control siblings (D) and *scospondin*^icm15/icm15^ mutants (E). Data were collected from three independent experiments and include 9, 11, and 12 control embryos recorded after vehicle, epinephrine and norepinephrine injections respectively (n = 131, 150, and 164 cells respectively) and 11, 10, and 10 *scospondin*^icm15/icm15^ mutants after vehicle, epinephrine and norepinephrine injections respectively (n = 168, 124, and 150 cells respectively). Each point represents a single cell. Bottom and top edges of the boxes indicate the 1st and 3rd quartiles. Dotted lines represent the distribution range around the 1st and 3rd quartiles of control embryos injected with a vehicle solution. Bold lines represent the median value for each distribution. ns p>0.05, ***p<0.001 (Kolmogorov-Smirnov test). See also *Figure 5—figure supplement 1* and *Figure 5-video 1*.

The online version of this article includes the following video, source data, and figure supplement(s) for figure 5:

**Source data 1.** Data for *Figure 5D–E*.

**Figure supplement 1.** Exogenous norepinephrine injected in brain ventricles is transported to the spinal cord and saturates the central canal 60 min post-injection.

**Figure 5—video 1.** Epinephrine and norepinephrine restore intracellular calcium transients of ventral CSF-contacting neurons in *scospondin* mutants.
https://elifesciences.org/articles/59469#fig5video1

the effect of epinephrine and norepinephrine in *scospondin*<sup>icm15/icm15</sup> mutants (*Figure 5C and E*, *Figure 5-video 1*). As described previously, ventral CSF-cNs displayed a 44.1% decrease of calcium variations in *scospondin*<sup>icm15/icm15</sup> embryos compared to control siblings in control condition with vehicle injections (median value; *Figure 5D and E*), confirming the observations reported in *Figure 3*. Interestingly, injections of epinephrine or norepinephrine in *scospondin*<sup>icm15/icm15</sup> mutants increased the median $\int \Delta F/F$ min$^{-1}$ by 70.02 and 88.6%, respectively compared to vehicle injections (*Figure 5E*). Moreover, the rescue of the spontaneous activity of CSF-cNs by monoamines in mutant embryos reached comparable levels to those observed in control siblings (*Figure 5D–E*). These data show that epinephrine and norepinephrine restore the Reissner fiber-dependent calcium transients in ventral CSF-cNs in *scospondin* mutants.

Altogether, our results are compatible with the existence of a signal that links the Reissner fiber to both *urp2* expression and calcium variations in ventral CSF-cNs that can be modulated by mono-aminergic activation. These observations suggest that endogenous epinephrine and norepinephrine may act locally to tune calcium signaling in CSF-cNs.

## Norepinephrine can be detected in the embryonic CSF and adrenergic receptors are expressed by spinal cells contacting the CSF

To assess that this modulation takes place at the interface between CSF and the cells lining the central canal, we performed immunostainings against endogenous norepinephrine at 30 hpf in *scospondin*<sup>ut24Tg</sup> embryos labeling in vivo the Reissner fiber with GFP (*Troutwine et al., 2020*). As previously described, SCO-spondin positive material in the CSF contributes mainly to form the fiber, but is also present as punctated material in close vicinity the fiber, referred to here as extrafibrillar material, (*Troutwine et al., 2020*, *Figure 6A*). We observed norepinephrine positive puncta colocalized with the Reissner fiber as well as endogenous norepinephrine positive signals following patterns similar to extrafibrillar material labeled by the SCO-spondin-GFP fusion protein. This can also be observed closely at the level of the *massa caudalis*, formed by the accumulation of SCO-spondin at the caudal end of the central canal (*Figure 6A*). These observations suggest that norepinephrine is endogenously present in the embryonic CSF where it is associated with the Reissner-positive material in the central canal.

To address the question of the receptor associated to this signaling, we performed immunostainings against the adrenergic receptor Adrb2, binding both epinephrine and norepinephrine, which is described to be transiently expressed in the zebrafish nervous system at early stages of embryonic life (*Wang et al., 2009*). We observed that Adrb2 is distributed ventrally in the neural tube, at the interface with the central canal, in a pattern that suggests a membrane localization in both control siblings and *scospondin*<sup>icm15/icm15</sup> mutants (*Figure 6B*). Thus, Adrb2 localization is suitable for binding endogenous ligands present in the CSF.

Next, we addressed the question of whether Adrb2 is expressed in CSF-contacting neurons, which could explain the rescue of calcium transients observed in *scospondin*<sup>icm15/icm15</sup> mutants after epinephrine and norepinephrine injections. Double immunostainings against Adrb2 and GFP in *Tg (pkd2l1:GCaMP5G)* control siblings show that both signals cover different domains in the ventral most region of the spinal cord (*Figure 6C*), where the Adrb2 positive domain remains in the ventral midline, inserted between two rows of ventral CSF-contacting neurons (*Figure 6—video 1*). As ventral CSF-contacting neurons are known to derive from lateral floor plate progenitors (*Park et al., 2004*; *Yang et al., 2010*), the medial cells expressing Adrb2 most probably correspond to the medial floor plate. Importantly, the same distribution is observed in *scospondin*<sup>icm15/icm15</sup> mutants (*Figure 6C* and *Figure 6—video 1*). This observation suggests that the loss of calcium signaling in *scospondin*<sup>icm15/icm15</sup> mutants is unlikely due to a defect in the Adrb2 distribution in the spinal cord.

Altogether, our results indicate that endogenous adrenergic signals could modulate the Reissner fiber-dependent signaling pathway that instructs body axis straightening during embryonic development, and suggests that monoamines act on CSF-contacting neurons through an indirect mechanism.

## Discussion

Using a combination of transcriptomic analyses together with in vivo calcium imaging and pharmacology, we show here that the Reissner fiber is essential for signaling to the developing CSF-

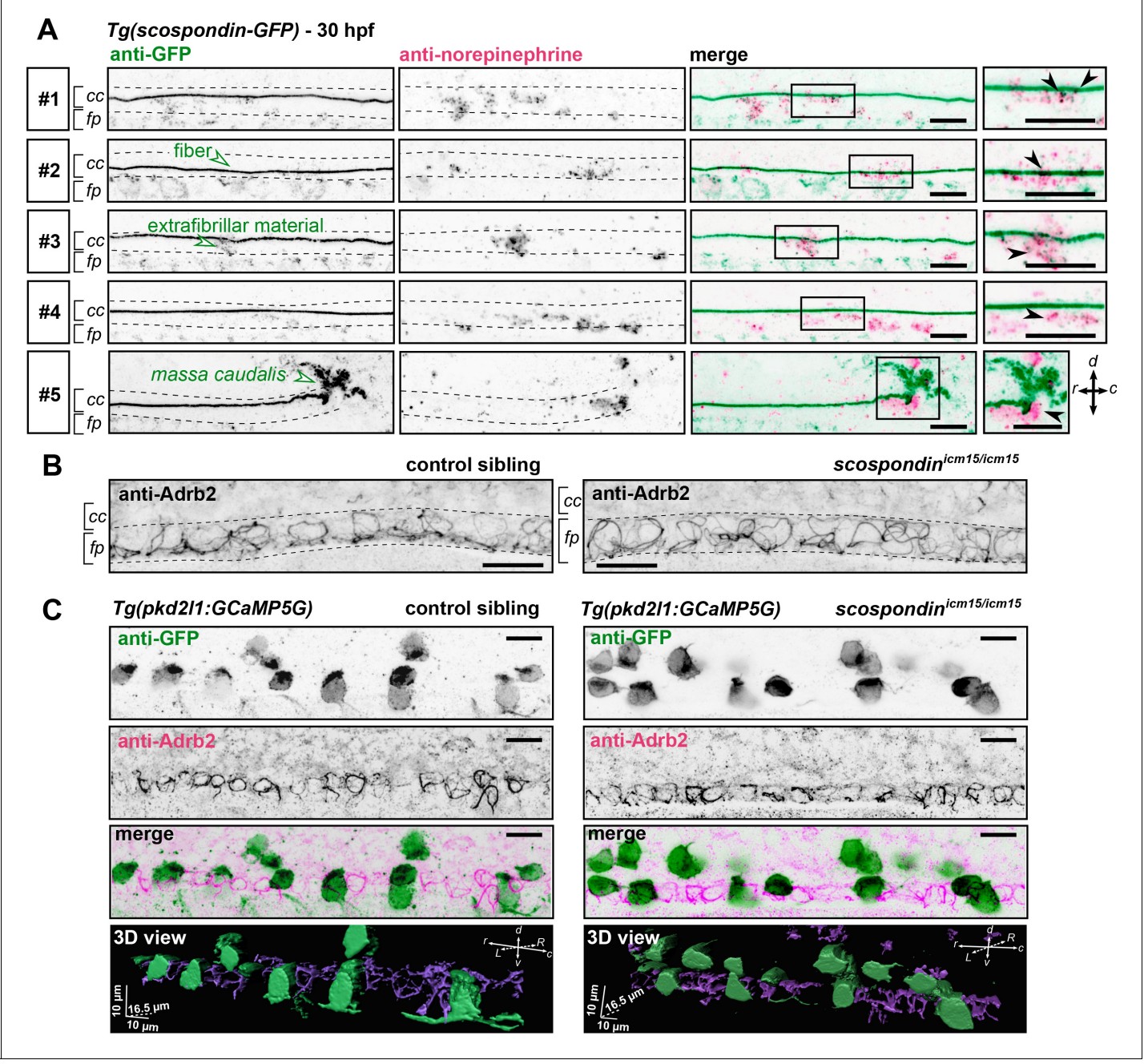

**Figure 6.** The adrenergic receptor Adrb2 is expressed in cells ventral to the central canal in which norepinephrine can be detected. (**A**) Double immunodetection of GFP (left) and endogenous norepinephrine (middle) in the spinal cord of *scospondin^ut24Tg^* embryos at 30 hpf imaged laterally. Merged signals and close up of boxed regions are represented on the right. 5 representative examples (#1 to #5) out of 26 embryos are shown (no signal was detected in 6 embryos out of 26 in total, n = 2 independent experiments). In the central canal, norepinephrine positive signals can be detected as colocalized with the Reissner fiber itself (black arrowheads; embryos #1, 2, 3), with extrafibrillar material in the central canal (black arrowheads; embryos #4 and #5), and closely apposed to the *massa caudalis* located after the caudal limit of the central canal (black arrowhead; embryo #6). Scale bars: 10 μm. r: rostral; c: caudal; d: dorsal; v: ventral; cc: central canal; fp: floor plate. (**B**) Immunohistochemistry for the adrenergic receptor Adrb2 in a 30 hpf control sibling (left, representative example out of 8 embryos) and a *scospondin^icm15/icm15^* embryo (right, representative example out of 9 embryos). Adrb2 is distributed along the midline of the ventral most region of the neural tube (corresponding to the floor plate, fp) at the interface with the central canal (cc). Embryos are oriented rostral to the left and dorsal to the top. Scale bars: 10 μm. (**C**) Sagittal views of double immunostainings against GFP and Adrb2 in the spinal cord of *Tg(pkd2l1:GCaMP5G)* embryos at 30 hpf. Maximal z-projections are shown for a control sibling (left; one embryo out of 12) and a *scospondin^icm15/icm15^* mutant (right; one embryo out of 8). Merged signals show that GFP-positive and Adrb2 positive signals cover different domains in the ventral most region of the neural tube. 3D reconstructions of the same field of views (bottom panel: GFP: *Figure 6 continued on next page*

*Figure 6 continued*

green; Adrb2: purple) further illustrate that Adrb2 is enriched in a distinct cell population that is medial to ventral CSF-contacting neurons. Scale bars: 10 μm. L: left; R: right; See also *Figure 6—video 1*.

The online version of this article includes the following video for figure 6:

**Figure 6—video 1.** Adrb2 is expressed in the midline of the ventral most region of the neural tube that is different from that of CSF-contacting neurons.

https://elifesciences.org/articles/59469#fig6video1

---

contacting neurons (CSF-cNs). In these neurons, the Reissner fiber is required for both *urp2* expression and spontaneous intracellular calcium variations. This functional interaction between the Reissner fiber and ventral CSF-cNs is required for a normal curvature of the developing posterior axis of zebrafish. Using monoamine injections into the CSF, we show that the CSF-cNs response to the signal from the Reissner fiber can be modulated by local adrenergic activation in vivo, suggesting that the Reissner fiber acts by controlling the availability of a chemical signal in the CSF.

## *urp2* expression in ventral CSF-cNs depends on the presence of the Reissner fiber and impacts on the curvature of embryonic axis

Previous results in both cilia-defective and *scospondin* mutants suggested that mutants that fail to form a Reissner fiber display a strong downregulation of both Urotensin-II-related neuropeptides 1 and 2 in the embryo (*Lu et al., 2020*; *Rose et al., 2020*; *Zhang et al., 2018*). Our new results contradict the observation that *urp1* is downregulated in *scospondin* mutants (*Lu et al., 2020*). In the present study, our transcriptomic analysis of *scospondin* mutants yielded surprisingly few candidates commonly misregulated in the two alleles, and *urp2* was the only gene with a strong downregulation, while *urp1* showed no significant change. We confirmed this result by qRT-PCR and showed again that only *urp2* is strongly downregulated, while *urp1* transcripts levels were not significantly affected. In line with our results, the investigation of a newly generated *scospondin* hypomorphic allele (*Rose et al., 2020*) also revealed *urp2* as the major downregulated gene in homozygous mutant and did not report a *urp1* downregulation.

We further show that *urp2* overexpression in *scospondin* mutants decreases the frequency of curled-down phenotypes, confirming that *urp2* expression level is involved in the control of embryonic axis curvature (*Lu et al., 2020*; *Zhang et al., 2018*). However, *urp2* may not be the only determinant of embryonic axis morphogenesis downstream of the Reissner fiber, as single *urp2* morpholino knockdown does not show a defective axis curvature (*Zhang et al., 2018*). Parallel signaling pathways that would likely act post-transcriptionally and would therefore not be detected using our transcriptomic strategy may be involved.

The discovery of a new hypomorphic allele for *scospondin* recently revealed an inflammatory signature induced by the loss of the Reissner fiber at the embryonic stage (*Rose et al., 2020*). In our work, we did not detect such a signature in our transcriptomic analysis (see *Figure 1—figure supplement 1*). We can speculate that these differences are due to difference in the fish genetic background or husbandry conditions. Our work therefore begs for further studies in order to decipher the molecular pathways downstream and/or in parallel to Urp2 that regulate the morphogenesis of the embryonic axis.

## Intracellular calcium variations in developing ventral CSF-cNs require the Reissner fiber

In the embryo, ventral CSF-cNs are spontaneously active via the opening of the Pkd2l1 calcium channel enriched at the level of the developing apical extension of the cells (*Sternberg et al., 2018*). In addition to controlling *urp2* gene expression in CSF-cNs, we report here that the Reissner fiber is also required for calcium signaling in *urp2*-expressing cells. By investigating the *pkd2l1* mutant deprived of calcium signaling in embryonic CSF-cNs, our work rejects the simple explanation that the Reissner fiber controls *urp2* mRNA level and axis straightness by increasing calcium intracellular concentrations in CSF-cNs. Indeed, the absence of *urp* genes downregulation observed in the *pkd2l1* mutant compared to wild-type goes against this hypothesis.

One question remaining is how the Reissner Fiber controls calcium variations in ventral CSF-cNs. We recently showed that the Reissner fiber is functionally coupled to the mechano-sensory function of these interoceptive neurons in the larva (*Orts-Del'Immagine et al., 2020*). However, embryonic CSF-cNs are not fully differentiated, as they do not harbor a fully developed apical extension known to tune their mechanosensory function at larval stage (*Desban et al., 2019*). We therefore favor the hypothesis that the Reissner fiber acts via the modulation of the CSF content, which is supported by the restoration of calcium signaling in CSF-cNs upon monoamine injections in the brain ventricles.

## Adrenergic activation restores the Reissner fiber-dependent signaling and axis straightening

Earlier reports suggest that the Reissner fiber can interact with different neuro- modulators, including monoamines (*Caprile et al., 2003*; *Diederen et al., 1983*). Monoamines globally supplied in the fish water were also found to rescue body curvature in Reissner fiber-defective mutants (*Lu et al., 2020*; *Zhang et al., 2018*). They are therefore good candidates to influence the Reissner fiber signal towards ventral CSF-cNs. We added to previous report (*Lu et al., 2020*) that the action of both epinephrine and norepinephrine can compensate locally, directly in the ventricular cavities, for the loss of the Reissner fiber on calcium signaling. We also used quantitative PCR to confirm the regulation of *urp* neuropeptides expression by both epinephrine and norepinephrine in *scospondin* mutants, extending the previous result obtained using in situ hybridization after epinephrine treatments (*Lu et al., 2020*). These two monoamines rescue the *scospondin* phenotype for three features: spontaneous calcium variations and Urotensin-II-related peptides expression in CSF-cNs, and body axis curvature.

One possible interpretation of our results is that the Reissner fiber is essential for the control of endogenous epinephrine and norepinephrine distribution in the embryonic CSF. This is consistent with our immunodetection of norepinephrine in close vicinity with the Reissner fiber in the embryonic central canal. This hypothesis is supported by the presence of noradrenergic neurons in the embryonic hindbrain as early as 24 hpf (*Holzschuh et al., 2003*), providing a potential source of monoamines that would need to be transported caudally to the central canal of the spinal cord. This hypothesis is reinforced by our original result showing the localization of the Adrb2 receptor at the interface with CSF in the ventral part of the neural tube, ideally located to bind norepinephrine ligands that we found distributed close to Reissner-positive material. Interestingly Adrb2 belongs to the G protein-coupled receptors family and was identified to trigger cytoplasmic calcium raise in vitro (*Galaz-Montoya et al., 2017*). In zebrafish, the adrenergic system plays important roles in the control of wakefulness (*Singh et al., 2015*) and of cardiac contractions (*Steele et al., 2011*). However, no morphological defects have been reported in animal missing the rate-limiting enzyme for the synthesis of epinephrine and norepinephrine (*Singh et al., 2015*). This might reflect the masking of the phenotype due to transcriptional adaptation to the genetic mutation (*Rossi et al., 2015*). Alternatively, redundant signaling pathways might mask the role of monoamines in axis curvature control. Nonetheless, our results support the idea that the cross-talk between the Reissner fiber and undifferentiated CSF-cNs is likely to be of chemical nature, possibly through monoamines themselves. They further support the idea that monoamines act on CSF-cNs in an indirect manner. Future investigations will allow to fully delineate the contribution and mechanism of action of endogenous monoamines on ventral CSF-cNs signaling and body axis curvature.

Altogether, our study unravels a signal from the Reissner fiber to the developing CSF-contacting neurons. We also show that adrenergic activation can modulate this signal during embryonic body axis morphogenesis. Interestingly, a temporally controlled inactivation of cilia motility leads to spine curves reminiscent of adolescent idiopathic scoliosis (*Grimes et al., 2016*). Recent results suggest that, as in the embryo, the Reissner fiber and Urp reception in slow muscles are also implicated in the maintenance of a straight spine during postembryonic development (*Lu et al., 2020*; *Rose et al., 2020*; *Troutwine et al., 2020*; *Zhang et al., 2018*). Our work will pave the way for future investigations to identify the potential of the interplay between CSF-cNs and the adrenergic system to modulate the Reissner fiber-dependent morphogenesis of the spine in the juvenile.

# Materials and methods

## Key resources table

| Reagent type (species) or resource | Designation | Source or reference | Identifiers | Additional information |
|---|---|---|---|---|
| Genetic reagent (*D. rerio*) | *scospondin*[icm13] | *Cantaut-Belarif et al., 2018* | ZFIN : ZDB-ALT-181113–3 | |
| Genetic reagent (*D. rerio*) | *scospondin*[icm15] | *Cantaut-Belarif et al., 2018* | ZFIN: ZDB-ALT- 181113–4 | |
| Genetic reagent (*D. rerio*) | *pkd2l1*[icm02] | *Böhm et al., 2016* | ZFIN : ZDB-ALT-160119–6 | |
| Genetic reagent (*D. rerio*) | *Tg(pkd2l1:GCaMP5G)*[icm07Tg] | *Böhm et al., 2016* | ZFIN: ZDB-ALT-160119–4 | |
| Genetic reagent (*D. rerio*) | *Tg(pkd2l1:GAL4)*[icm10Tg] | *Fidelin et al., 2015* | ZFIN: ZBD-ALT-150324–1 | |
| Genetic reagent (*D. rerio*) | *Tg(UAS:mCherry)* | *Robles et al., 2014* | ZFIN: ZBD-ALT-130702–1 | |
| Genetic reagent (*D. rerio*) | *scospondin*[ut24Tg] | *Troutwine et al., 2020* | | R. S. Gray lab |
| Recombinant DNA reagent | *pCS2-urp2* plasmid | G. Pézeron, this work | | Used for RNA synthesis |
| Recombinant DNA reagent | *pCS2-Ras-eGFP* plasmid | *Ségalen et al., 2010* | | Used for RNA synthesis |
| Sequence-based reagent | URP2_BamHI_F gcgcgcGGATCC gtatctgtagaatctgctttgctgc | This work | | forward oligonucleotide used for *urp2* cloning |
| Sequence-based reagent | URP2_XbaI_R gcgcgcTCTAGA ggcagagggtcagtcgtgttat | This work | | reverse oligonucleotide used for *urp2* cloning |
| Sequence-based reagent | *urp2* forward: CCACCGGATCACCATCATTACC | This work | | qPCR oligonucleotide |
| Sequence-based reagent | *urp2* reverse: GATGCCACCGCTGTCTATAGTG | This work | | qPCR oligonucleotide |
| Sequence-based reagent | *urp1* forward: TGCGCTGCCTCTGTATTCAG | This work | | qPCR oligonucleotide |
| Sequence-based reagent | *urp1* reverse: CTTTGTCCGTCTTCAACCTCTG | This work | | qPCR oligonucleotide |
| Sequence-based reagent | *pkd2l1* forward: GCGAACTATGCCCAATGAGG | This work | | qPCR oligonucleotide |
| Sequence-based reagent | *pkd2l1* reverse: TCTCAAAGCTGTTCCCCACA | This work | | qPCR oligonucleotide |
| Sequence-based reagent | *lsm12b* forward: GAGACTCCTCCTCCTCTAGCAT | This work | | qPCR oligonucleotide |
| Sequence-based reagent | *lsm12b* reverse: GATTGCATAGGCTTGGGACAAC | This work | | qPCR oligonucleotide |
| Antibody | Anti-Reissner fiber, rabbit, polyclonal | *Didier et al., 1995* | Courtesy of S. Gobron | Dilution 1:200 |
| Antibody | Anti-GFP, chicken, polyclonal | Abcam, Cat# ab13970 | RRID:AB_300798 | Dilution 1:500 |
| Antibody | Anti-Pkd2l1, rabbit, polyclonal | *Sternberg et al., 2018* | | Dilution 1:200 |
| Antibody | Anti-norepinephrine, rabbit, polyclonal | Millipore, Cat# AB120 | RRID:AB_90481 | Dilution 1:100 |
| Antibody | Anti-Adrb2, rabbit, polyclonal | ThermoFischer Scientific, Cat# PA5-80323 | RRID:AB_2787652 | Dilution 1:200 |
| Antibody | Alexa Fluor-488 goat anti chicken IgG (H+L) | Molecular Probes, Cat# A-11039 | RRID:AB_142924 | Dilution 1:500 |
| Antibody | Alexa Fluor-568 goat anti-rabbit IgG (H+L) | Molecular Probes, Cat# A-11036 | RRID:AB_10563566 | Dilution 1:500 |
| Antibody | Alexa Fluor-488 donkey anti rabbit IgG (H+L) | Molecular Probes, Cat# A-21206 | RRID:AB_2535792 | Dilution 1:500 |
| Chemical compound, drug | DL-Norepinephrine hydrochloride | Sigma | Cat# A7256 | Dilution to 3 mM |
| Chemical compound, drug | +- Epinephrine hydrochloride | Sigma | Cat# E4642 | Dilution to 3 mM |

*Continued on next page*

*Continued*

| Reagent type (species) or resource | Designation | Source or reference | Identifiers | Additional information |
|---|---|---|---|---|
| Chemical compound, drug | MS 222 | Sigma | Cat# E10521 | Dilution to 0.2% w/v |
| Chemical compound, drug | alpha-bungarotoxin | Tocris | Cat# 2133 | Dilution to 500 µM |
| Software, algorithm | MATLAB | MathWorks | RRID:SCR_001622 | |
| Software, algorithm | Prism | GraphPad | RRID:SCR_002798 | |
| Software, algorithm | Fiji | *Schindelin et al., 2012* | RRID:SCR_002285 | |
| Software, algorithm | Imaris | Oxford Instruments | RRID:SCR_007370 | |

## Animal husbandry and genotyping

All procedures were performed on zebrafish embryos between 30 and 48 hpf in accordance with the European Communities Council Directive (2010/63/EU) and French law (87/848) and approved by the Paris Brain Institute (Institut du Cerveau) and the French Ministery for research (APAFIS agreement 2018071217081175). All experiments were performed on *Danio rerio* embryos of AB, Tüpfel long fin (TL) and nacre background. Animals were raised at 28.5℃ under a 14/10 light/dark cycle until the start of the experiment. Genotyping was performed as previously described for *scospondin*[icm13] and *scospondin*[icm15] (*Cantaut-Belarif et al., 2018*) and *pkd2l1*[icm02] (*Böhm et al., 2016*).

## RNA sample preparation

For each experiment, condition (developmental time point and pharmacological treatment) and phenotype (either straight or curled-down), 30 sibling embryos were sorted in independent vials. After euthanasia in 0.2% MS 222 (Sigma, E10521), embryos were resuspended in 1 mL of Trizol (Thermo-Fischer Scientific, 15596026) and dissociated by multiple aspirations through the needle of a syringe. 200 µL chloroform were added prior to centrifugation and extraction of the aqueous phase. Nucleic acids were precipitated using 700 µL isopropanol, and the pellet was resuspended in 200 µL water. RNAs were purified using the RNeasy Micro Kit (Qiagen, 74004), following the provider's instruction. We performed the optional on-column DNAse treatment to improve RNAs purity. RNAs were eluted in 30 µL water to ensure a high concentration, and the quality and quantity of the extract were evaluated on a TapeStation System (Agilent).

RNA sequencing analysis mRNA library preparation was realized following manufacturer's recommendations (KAPA mRNA HyperPrep Kit, Roche). Final samples pooled library preparations were sequenced on Nextseq 500 ILLUMINA, corresponding to $2 \times 30$ millions of 75 base pair reads per sample after demultiplexing. Quality of raw data was evaluated with FastQC (*Andrews, 2010*). Poor quality sequences were trimmed or removed using Trimmomatic software (*Bolger et al., 2014*) to retain only good quality paired reads. Star v2.5.3a (*Dobin et al., 2013*) was used to align reads on GRCz11 reference genome using standard options. Quantification of gene and isoform abundances was achieved using with Rsem 1.2.28, prior to normalization with the edgeR bioconductor package (*Robinson et al., 2010*). Finally, differential analysis was conducted with the GLM framework likelihood ratio test from edgeR.

The quality control and PCA analysis (not shown) indicated that one sample was not reaching the same reproducibility as the two others, even after trimming and normalization. We therefore used the average value of the two most reliable replicates to calculate the average expression level (count per million, cpm) of all the 28,214 genes either in straight controls or in curled-down embryos (average of the expression in the two alleles *icm13* and *icm15*). We then filtered out low-expression genes that had an expression level below 1 c.p.m. in the average of the straight control replicates. We then kept genes that had an average fold change between curled embryos and controls of 0.75 for down-regulated genes (120 genes), and 1.45 for up-regulated genes (94 genes). We turned back to the original raw cpm values of the three replicates in the two alleles for these two short-lists. We tested the consistency of the fold change across replicates and across alleles with a general linear model (GLM) (*Nelder and Wedderburn, 1972*). The design matrix included 2 regressors of interest (encoding whether the sample is a straight control or curled embryos) and two confounding variables (the unwanted variability that might be associated with the two different genetic environments

of the two families of fish carrying the *icm13* and the *icm15* allele). Statistical significance of the effect of interest (above and beyond confounding factors) was tested using a t-test. Because of the potential false discovery associated with multiple testing, we then used the Benjamini–Hochberg procedure (*Benjamini and Hochberg, 1995*).

Potential genes of interest were sorted according to their increasing p-values. To shorten the list of potential genes of interest, we determined at which rank i the p-value became higher than the Benjamini–Hochberg criterion using (i/total nb)$\times$ 0.2. This method allows for up to 20% of false-positives, but avoids the rejection of true-positives that was manifest with a more stringent correction as the Bonferroni correction. For example, the *scospondin* mRNA level in the *icm13* allele is highly decreased, presumably through non-sense mediated decay: the Bonferroni correction would have rejected this result as not significant, whereas the Benjamini–Hochberg procedure keeps it in the list of potentially interesting genes (see *Figure 1—source data 1* for the complete lists). However, this procedure requires a post-hoc validation with an independent technique for each potential gene of interest, as we did by qRT-PCR (see below).

## Quantitative RT-PCR

1.2 µg of each RNA sample was retro-transcribed using the Verso cDNA Synthesis Kit (Thermo-Fischer Scientific, AB1453B) following the provider's instructions. A 1:3 ratio of random hexamer:polydT primers was used to favor mRNA amplification without a strong bias for the 3' end of messengers. qPCR experiments were performed using the LightCycler 480 SYBR Green I Master kit (Roche, 04707516001) on a LightCycler 96 machine (Roche). Each pair of primers (see Key Resource Table) was tested beforehand on a given cDNA stock that was diluted in a series of 4 points. Linearity of CT variation with the cDNA dilution as well as single peaks in the melting curves corresponding to single amplicons were assessed. For each new RNA extraction, we tested that no amplification was detectable when the PCR was performed directly on the RNA stock (-RT). We used the housekeeping gene *lsm12b* as an internal reference in each experiment. qPCRs were repeated two to three times for each cDNA stock and the average CT was used for further calculations. The relative abundance of the gene of interest was evaluated using the CT comparison formula: $2e^{-\Delta CT}$. All results were obtained on at least three biological replicates (except for *pkd2l1* mutant extracts), each originating from a single mating.

## Monitoring of body axis curvature in developing embryos

To characterize the exact evolution of the geometry of the posterior axis in *scospondin*^icm15/icm15 mutants during the early stages on embryogenesis (*Figure 1—figure supplement 1*), we performed incrosses of *scospondin*^icm15/+ parents. Eggs were collected 30 min after mating and maintained at 28.5°C until 19.5 hpf. Embryos were staged according to *Kimmel et al., 1995* at 19.5 hpf. Individual embryos were mounted laterally at 20 hpf in 1.5% low-melting point agarose, imaged using a AZ100M macroscope (Nikon), unmounted and kept at 28.5°C until they were remounted for the next stages (22, 24, 28, and 30 hpf). The time window necessary for mounting-imaging-unmounting at each step was short enough (10–15 min) to be neglected.

## Pharmacology and quantification of body axis curvature

30 hpf embryos from *scospondin*^icm15/+ incrosses were treated during 2.5 hr at 28.5°C with E3 medium alone (vehicle), epinephrine hydrochloride (Sigma, E4642) or norepinephrine hydrochloride (Sigma, A7256) both diluted to 3 mM in E3 medium (*Figure 4*). To avoid light-induced oxidation of epinephrine and norepinephrine, dishes were covered with foil paper during the incubation time. To ensure a proper quantification of ear-to-tail angles, embryos were then fixed over-night in 4% PFA 6-well plates, rinsed 3 times during 45 min in 1X PBS, mounted laterally in 1.5% low-melting point agarose and imaged using a AZ100M macroscope (Nikon). For each embryo, the angle between the ear, the caudal limit of the yolk extension and the tip of the tail (*Figure 4C*) was quantified using Fiji (*Schindelin et al., 2012*). Representative traces of the global morphology of the embryos after treatment were drawn for all experimental conditions on one representative clutch (*Figure 4B*) by linking the center of the eye to the ear and following the dorsal line linking somite boundaries until the tip of the tail.

## In vivo calcium imaging

*Tg(pkd2l1:GCaMP5G)* embryos were manually dechorionated at 30 hpf, mounted laterally in 1.5% low-melting point agarose, and paralyzed by injecting 1–2 nL of 500 µM alpha-bungarotoxin (Tocris, 2133) in the caudal muscles of the trunk. When required (*Figure 5*), hindbrain ventricle injections were performed using artificial CSF (aCSF, containing in mM: 134 NaCl, 2.9 KCl, 1.2 MgCl$_2$, 10 HEPES, 10 glucose, and 2.1 CaCl$_2$; 290 mOsm.kg$^{-1}$, adjusted to pH 7.7–7.8 with NaOH) as a vehicle solution. Epinephrine hydrochloride (Sigma, E4642) and norepinephrine hydrochloride (Sigma, A7256) were diluted to 3 mM in aCSF before hindbrain ventricle injections, using a sharp funnel shape needle with an approximate tip diameter of 1–3 µm. Embryos were imaged 20 to 60 min after hindbrain ventricle injections. Calcium imaging was performed at 4 Hz using a spinning disk confocal microscope (Intelligent Imaging Systems, Denver) for 4 min. Imaging was restricted to the region of the spinal cord located above the yolk extension. Regions of interest were manually selected based on an average projection of the time-lapse and according to the dorso-ventral position of the cells. The integrals of normalized ΔF/F signals were calculated using a custom script on MATLAB (see full description in *Sternberg et al., 2018*). The MATLAB codes used for data analysis are available at Github https://github.com/wyartlab/Cantaut-Belarif-et-al.-2020 (*Cantaut-Belarif et al., 2020*; copy archived at swh:1:rev:3396034de4726cb8c895a6e43bbc3d774b726fcb).

## In vivo patch clamp recording

Whole-cell recordings were performed in aCSF on 30 hpf double transgenic *Tg(pkd2l1:GAL4;UAS: mCherry)* embryos carrying the *scospondin$^{icm15}$* mutation and their respective control siblings. Embryos were pinned through the notochord with 0.025 mm diameter tungsten pins. Skin and muscle from two to three segments around segment 10 were dissected out using a glass suction pipette. A MultiClamp 700B amplifier, a Digidata series 1440 A Digitizer, and pClamp 10.3 software (Axon Instruments, Molecular Devices, San Jose, California, USA) were used for acquisition. Raw signals were acquired at 50 kHz and low-pass filtered at 10 kHz. Patch pipettes (1B150F-4, WPI) with a tip resistance of 6–8 MΩ were filled with internal solution containing in (mM: K-gluconate 115, KCl 15, MgCl2 2, Mg-ATP 4, HEPES-free acid 10, EGTA 5 or 10, 290 mOsm/L, pH adjusted to 7.2 with KOH with Alexa 488 at 40 µM final concentration). Holding potential was − 85 mV, away from the calculated chloride reversal potential (E$_{Cl}$ = - 51 mV).

## Immunohistochemistry and confocal imaging

Embryos were manually dechorionated and euthanized using 0.2% MS 222 (Sigma, E10521) prior to fixation.

To detect the Pkd2l1 channel together with GFP (*Figure 2D*), 30 hpf embryos were fixed 4 hr at 4˚C in 4% PFA and then washed 3 times during 30 min in 1X PBS. Embryos were blocked overnight at 4˚C in a solution containing 0.7% Triton, 1% DMSO, and 10% NGS. The primary antibodies were incubated overnight at 4˚C in a solution containing 0.5% Triton, 1% DMSO, and 1% NGS and subsequently washed four times during 1 hr in a solution containing 0.5% Triton, and 1% DMSO (washing solution). Secondary antibodies were incubated 3 hr at room temperature in a solution containing 0.5% Triton, 1% DMSO, and 1% NGS. The mix of secondary antibodies was centrifuged 10 min at 10 000 rpm prior to the incubation to pellet unconjugated dyes (supernatant only was used for the incubation). Embryos were then washed 4 times during 2 hr using the washing solution and thereafter overnight at 4˚C in 1X PBS.

To detect the RF (*Figure 3C*), 30 hpf embryos were fixed 4 hr at 4˚C in 4% PFA and washed three times during 30 min in 1X PBS. Embryos were then blocked overnight in a solution containing 0.5% Triton, 1% DMSO, and 10% NGS. The primary antibody was diluted in a solution containing 0.5% Triton, 1% DMSO, and 1% NGS and incubated overnight at 4˚C. Embryos were subsequently washed 4 times during 1 hr in a solution containing 0.5% Triton, and 1% DMSO (washing solution). Secondary antibodies were incubated 3 hr at room temperature in a solution containing 0.5% Triton, 1% DMSO and 1% NGS. The mix of secondary antibodies was centrifuged 10 min at 10 000 rpm prior to the incubation to pellet unconjugated dyes (supernatant only was used for the incubation). Embryos were subsequently washed 4 times during 2 hr using the washing solution and thereafter overnight at 4˚C in 1X PBS.

To detect exogenous (*Figure 5—figure supplement 1*) and endogenous (*Figure 5A*) norepinephrine alone (*Figure 5—figure supplement 1*) or together with GFP (*Figure 5A*), we used an antibody designed against a Glutaraldehyde-conjugated form of norepinephrine (Millipore, AB120). Thus, 30 hpf embryos were fixed 1.5 hr at 4°C under agitation in a 4% PFA-0.125% Glutaraldehyde-3% sucrose solution to allow reactivity. Embryos were then washed 3 times in 1X PBS during 30 min and the yolks and skin from rostral parts of the trunk were removed. Embryos were blocked overnight at 4°C in a solution containing 0.6% Triton, 1.2% DMSO, and 10% normal goat serum (NGS). An additional blocking step was performed during 3 hr at room temperature the day after. Primary antibodies were incubated in a solution containing 1% Triton, 1% DMSO, and 1% NGS during one day at room temperature plus one night at 4°C. Embryos were then washed extensively 4 times during 2 hr in a solution containing 0.5% Triton and 1% DMSO (washing solution). Secondary antibodies were incubated 3 hr at room temperature in a solution containing 1% Triton, 1% DMSO, and 1% NGS. The mix of secondary antibodies was centrifuged 10 min at 10 000 rpm prior to the incubation to pellet unconjugated dyes (supernatant only was used for the incubation). Embryos were subsequently washed 4 times during 2 hr using the washing solution, 4 times during 1 hr using a solution of 50% washing solution-50% 1X PBS and thereafter overnight at 4°C in 1X PBS.

To detect Adrb2 alone (*Figure 6B*) or together with GFP (*Figure 6C*), embryos were fixed 2 hr at 4°C in a 4% PFA-3% sucrose solution and washed 3 times in 1X PBS during 30 min. The yolks and the skin from the rostral part of the trunk were removed, and embryos were blocked overnight in a solution containing 0.6% Triton, 1.2% DMSO, and 10% normal goat serum (NGS). Primary antibodies were incubated at 4°C during 2 days in a solution containing 1% Triton, 1% DMSO, and 1% NGS. Embryos were washed extensively 4 times during 2 hr in a solution containing 0.5% Triton and 1% DMSO (washing solution). Secondary antibodies were incubated 3 hr at room temperature in a solution containing 1% Triton, 1% DMSO, and 1% NGS. The mix of secondary antibodies was centrifuged 10 min at 10 000 rpm prior to the incubation to pellet unconjugated dyes (supernatant only was used for the incubation). Embryos were subsequently washed using the washing solution 4 times during 2 hr, 4 times during 1 hr using a solution of 50% washing solution-50% 1X PBS and thereafter washed in 1X PBS overnight at 4°C.

The following dilutions of primary antibodies were used: rabbit anti-Reissner fiber 1:200 (*Didier et al., 1995*), rabbit anti-Pkd2l1 1:200 (*Sternberg et al., 2018*), chicken anti-GFP 1:500 (Abcam ab139170), rabbit anti-norepinephrine 1:100 (Millipore, AB120), rabbit anti-Adrb2 1:200 (ThermoFischer Scientific, PA5-80323). All secondary antibodies were from Molecular Probes and used at 1:500. Systematic omission of the primary antibody confirmed the specificity of the immunostaining results. Zebrafish embryos were mounted laterally in Vectashield Antifade Mounting Medium (Clinisciences, H1000) and imaged on an inverted SP8 DLS confocal microscope (Leica). Images were then processed using Fiji (*Schindelin et al., 2012*). Maximal Z-projections of 6–9 microns in depth are represented in *Figure 2D* and *Figures 3C*, 9-12 microns in depth in *Figure 6A and B* and Figure Supplement 4B and 18-22 microns in *Figure 6C*. 3D views shown in *Figure 6C* were obtained using Imaris.

## RNA microinjections

To produce mRNA, *urp2* CDS was amplified from cDNA by PCR and cloned (BamHI - XbaI) into pCS2+. Messenger RNA were produced with the mMESSAGE mMACHINE kit (Ambion). 1 nL RNA-containing solution was injected into 1- to 2 cell stage embryos obtained from *scospondin$^{icm15/+}$* incrosses. Each clutch was separated into three groups: uninjected and injected either with a control mRNA (100 ng / µL, *ras-eGFP* encoding for a membrane tagged GFP, [*Ségalen et al., 2010*]) or with a mix containing control mRNA and *urp2* mRNA (100 ng / µL total, 1:1 ratio). To assess for injection quality, GFP-positive embryos were first sorted out at 1 dpf and then scored at 48 hpf for body axis curvature defects.

## Statistics

All values are represented as boxplots (median ± interquartile range) or mean ± SEM (stated for each in the figure legend). All statistics were performed using MATLAB and Excel. In the figure panels, asterisks denote the statistical significance calculated using the appropriate test (stated for each test in the legends): *p<0.05; **p<0.01; ***p<0.001; ns, p>0.05.

## Acknowledgements

We gratefully thank Prof. Ryan S Gray for providing us with *Tg(scospondin-GFP)* embryos, Prof. Sylvie Schneider-Maunoury and Dr. Christine Vesque for providing us the Ras-eGFP plasmid, Dr. Stephane Gobron for the antibody directed against the Reissner fiber. We thank Monica Dicu and Antoine Arneau for fish care, the ICM.Quant imaging facility for instrument use, scientific and technical assistance and François-Xavier Lejeune for advice on statistics. This work benefited from equipment and services from the iGenSeq (RNA sequencing) and iCONICS (RNAseq analysis) core facilities at the ICM (Institut du Cerveau, Hôpital Pitié-Salpétrière, PARIS, France). We thank for critical feedback all members of the Wyart lab (www.wyartlab.com). This work was supported by, the HFSP Program Grants #RGP0063/2018, the Fondation Schlumberger pour l'Education et la Recherche (FSER/2017) for CW, and the Fondation des Treilles for YCB. The research leading to these results has received funding from the program 'Investissements d'avenir' ANR-10- IAIHU-06 (Big Brain Theory ICM Program) and ANR-11-INBS-0011 – NeurATRIS: Translational Research Infrastructure for Biotherapies in Neurosciences.

## Additional information

### Competing interests

Claire Wyart: Reviewing editor, *eLife*. The other authors declare that no competing interests exist.

### Funding

| Funder | Grant reference number | Author |
|---|---|---|
| Agence Nationale de la Recherche | ANR-10- IAIHU-06 | Yasmine Cantaut-Belarif<br>Margot Penru<br>Claire Wyart<br>Pierre-Luc Bardet |
| Agence Nationale de la Recherche | ANR-11-INBS-0011 | Yasmine Cantaut-Belarif<br>Adeline Orts Del'Immagine<br>Margot Penru<br>Claire Wyart<br>Pierre-Luc Bardet |
| Schlumberger Foundation | FSER/2017 | Claire Wyart |
| HFSP | #RGP0063/2018 | Yasmine Cantaut-Belarif<br>Claire Wyart |
| Fondation des Treilles | | Yasmine Cantaut-Belarif |
| New York Stem Cell Foundation | NYSCF-R-NI39 | Adeline Orts Del'Immagine<br>Claire Wyart |

The funders had no role in study design, data collection and interpretation, or the decision to submit the work for publication.

### Author contributions

Yasmine Cantaut-Belarif, Conceptualization, Data curation, Formal analysis, Investigation, Visualization, Methodology, Writing - original draft, Writing - review and editing; Adeline Orts Del'Immagine, Formal analysis, Investigation, Writing - review and editing; Margot Penru, Formal analysis, Investigation; Guillaume Pézeron, Resources, Methodology, Writing - review and editing; Claire Wyart, Conceptualization, Supervision, Funding acquisition, Validation, Writing - original draft, Writing - review and editing; Pierre-Luc Bardet, Conceptualization, Data curation, Formal analysis, Supervision, Funding acquisition, Validation, Investigation, Writing - original draft, Writing - review and editing

### Author ORCIDs

Yasmine Cantaut-Belarif https://orcid.org/0000-0002-8667-1483
Guillaume Pézeron http://orcid.org/0000-0003-1395-6397

Claire Wyart https://orcid.org/0000-0002-1668-4975
Pierre-Luc Bardet https://orcid.org/0000-0003-1766-1318

### Ethics

Animal experimentation: All procedures were performed on zebrafish embryos in accordance with the European Communities Council Directive (2010/63/EU) and French law (87/848). This project is included the APAFIS project #16469-2018071217081175 approved by the French Ministry for Research for the Paris Brain Institute (ICM).

### Decision letter and Author response

Decision letter https://doi.org/10.7554/eLife.59469.sa1
Author response https://doi.org/10.7554/eLife.59469.sa2

## Additional files

### Supplementary files

- Transparent reporting form

### Data availability

Data generated or analysed during this study are included in the manuscript and supporting files. Source data files have been provided for 5 figures. The raw RNA-seq data have been deposited in the ArrayExpress database at EMBL-EBI (www.ebi.ac.uk/arrayexpress) under accession number E-MTAB-9615.

The following dataset was generated:

| Author(s) | Year | Dataset title | Dataset URL | Database and Identifier |
|---|---|---|---|---|
| Cantaut-Belarif Y, Wyart C, Bardet PL | 2020 | scospondin mutant embryos | https://www.ebi.ac.uk/arrayexpress/experiments/E-MTAB-9615/ | ArrayExpress, E-MTAB-9615 |

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
