## [Decision Letter]

**Acceptance summary:**

In the manuscript by Cantaut-Belarif et al. the authors investigate how the Reissner fiber (RF) signals to the CSF neurons (CSFcNs). Using transcriptomics, state-of-the-art live imaging and in vivo electrophysiological analyses, they show that the RF elicits calcium signaling and controls the expression of *urp2* via local adrenergic signaling. They also show that local adrenergic signaling can rescue RF defects. Thus, the authors show that monoamines are mediators of the RF- CSF pathway that controls axis wide patterning of the embryo.

This is an interesting, well developed and clearly written manuscript that presents research conducted in parallel and independently from another report recently published by Lu et al. (Lu et al., 2020). While both manuscripts report similar developmental phenotypes, Cantaut-Belarif et al. present an in-depth analysis of monoamine signaling and its effects in CSFcNs that provides new mechanistic insights into the RF- CSF pathway.

**Decision letter after peer review:**

Thank you for submitting your article "Adrenergic activation modulates the Reissner fiber signal to cerebrospinal fluid-contacting neurons during development" for consideration by *eLife*. Your article has been reviewed by two peer reviewers, and the evaluation has been overseen by a Reviewing Editor and Didier Stainier as the Senior Editor. The reviewers have opted to remain anonymous.

The reviewers have discussed the reviews with one another and the Reviewing Editor has drafted this decision to help you prepare a revised submission.

This is an interesting and well written manuscript that investigates how the Reissner fiber (RF) signals to the CSF neurons (CSFcNs). The authors show that the RF elicits calcium signaling and controls the expression of *urp2* likely via adrenergic signaling, which can rescue RF defects. Importantly, the authors show that monoamines are mediators of the RF-CSF pathway that controls axis wide patterning of the embryo. There are however several points that need to be addressed experimentally before the manuscript is ready for publication.

Essential revisions:

1) Hindbrain injection of monoamines. It is unclear what concentrations were used, how long the monoamines were thought to be bioactive and how this was assessed, and to what extent they diffused. The authors should provide controls and validate they assay. Clear and detailed protocols are needed as well.

2) Double immunostaining with anti-GFP and anti-norepinephrine, and anti-Adrb2 in pkd2l1:GCAMP fish.

3) Justify the use of relatively late stage embryos (i.e. 30 hpf vs. 24 hpf). Provide images of *scospondin* mutants at the 20-26 hpf window.

4) Detailed protocols are needed.

5) Editorial revision should include comments on similarities and differences with Lu et al.

---

## [Author Response]

Essential revisions:1) Hindbrain injection of monoamines. It is unclear what concentrations were used, how long the monoamines were thought to be bioactive and how this was assessed, and to what extent they diffused. The authors should provide controls and validate they assay. Clear and detailed protocols are needed as well.

We agree with the reviewers that this is a critical point for the validation of the calcium imaging assay after monoamine hindbrain ventricle (HBV) injection in 30 hpf embryos.

First, concentrations of epinephrine and norepinephrine were already described in the Materials and methods section (epinephrine and norepinephrine were diluted to 3 mM in aCSF before hindbrain ventricle injections, see subsection “In vivo calcium imaging”). The time post-injection at which embryos were imaged was not mentioned and is now described in the revised version of the manuscript (embryos were imaged from 20 to 60 minutes post injection, see the Results subsection “Epinephrine and norepinephrine restore the Reissner fiber-dependent calcium signaling in ventral CSF-cNs of *scospondin* mutants”and in the Materials and methods subsection “In vivo calcium imaging”).

Second, we assessed experimentally whether norepinephrine delivered through HBV injection accessed the central canal of the spinal cord, was stable and present in excess in the CSF to trigger long-range signaling along the antero-posterior axis of the embryos. To address these points, we perfomed immunostainings against norepinephrine (no antibody is available to detect epinephrine) 30 and 60 minutes post HBV injection of norepinephrine diluted to 3mM in aCSF or aCSF only (“control injection”). The results are now depicted in a Figure 5—figure supplement 1. Our assay shows that norepinephrine-positive signal is present in the central canal 30 and 60 minutes post-injection in excess compared to control injections. These observations further confirms that norepinephrine injected in the HBV can diffuse down the central canal and is stable in the conditions used for in vivo calcium imaging. These results are described in the subsection “Epinephrine and norepinephrine restore the Reissner fiber-dependent calcium signaling in ventral CSF-cNs of *scospondin* mutants”. Note that our observations are in line with former studies using tritiated norepinephrine (^3^H-NE) injected into cerebral ventricles of dogs (Maas and Landis, 1965), sheep (Forbes and Bail, 1974) and rats (Fuxe and Ungerstedt, 1966, Levitt, Kowalik and Barkai, 1983) showing that ^3^H-NE is depleted 5 to 6 hours after the initial injection (see subsection “Epinephrine and norepinephrine restore the Reissner fiber-dependent calcium signaling in ventral CSF-cNs of *scospondin* mutants”).

2) Double immunostaining with anti-GFP and anti-norepinephrine, and anti-Adrb2 in pkd2l1:GCAMP fish.

We thank the reviewers for requesting important precisions here.

First, we assessed whether norepinephrine can be associated with the Reissner fiber and/or Reissner fiber-positive material in vivo at 30 hpf in the central canal of the spinal cord. We performed double immunostainings against norepinephrine (NE) and GFP in *Tg(scospondin-GFP)* embryos. We have extended our initial observation to 16 other embryos as shown in the new Figure 6A. Our results show that in 10 out of 16 embryos newly analyzed, NE-positive spots are also detected on the fiber itself or associated with GFP-positive deposits in the central canal. Moreover, NE-positive signals were also detected in close apposition to the *massa caudalis*, the structure formed by the aggregation of Reissner-positive material at the caudal limit of the central canal. To highlight these different co-localization or apposition patterns, we now show merged images of both signals in the novel Figure 6A exemplified on several embryos and further describe this result in the subsection “Norepinephrine can be detected in the embryonic CSF and adrenergic receptors are expressed by spinal cells contacting the CSF”.

Second, we specified the localization of the Adrb2 signal relative to CSF-cNs. To answer this question, we performed double immunostainings against Adrb2 and GFP on *Tg(pkd2l1:GCaMP5)* 30 hpf embryos (Figure 6B). Our results show that Adrb2 is not expressed in CSF-cNs but rather in more medial cells, which – according to their cuboidal shape and medial localization in the ventral part of the neural tube – likely correspond to the medial floor plate (see novel Figure 6B). This confirms that an adrenergic signal can be received in the spinal central canal, and suggests that monoamines act indirectly on CSF-cNs. Importantly, this expression pattern is conserved in *scospondin* mutants, suggesting that the defect in calcium signaling observed in CSF-cNs is unlikely due to a defect in the receptor localization in the absence of the RF (see the aforementioned subsection).

3) Justify the use of relatively late stage embryos (i.e. 30 hpf vs. 24 hpf). Provide images of scospondin mutants at the 20-26 hpf window.

As stated in our previous publication (Cantaut-Belarif et al., 2018), the curled-down phenotype becomes obvious in *scospondin* mutants at 28-30 hpf. We had noticed that *scospondin* mutants could not be phenotypically recognized earlier than this stage. This is the reason why 28-30 hpf and 48 hpf embryos were solely used to identify *scospondin* mutants according to their specific curled phenotype. The 28-30 hpf stage corresponds to a developmental time where calcium variations are very important in the wild-type ventral CSF-cNs (Sternberg et al., 2018). This is stated in the revised manuscript to justify our choice (subsections “The Reissner fiber controls *urp2* gene expression” and “The Reissner fiber is required for calcium signaling in *urp2* expressing CSF-cNs”).

In addition, to better illustrate the evolution of the posterior axis geometry between 20 and 30 hpf, we imaged the evolution of the posterior axis geometry over time on animals generated from incrosses of adult *scospondin^icm15/*^* parents. We provide now images of a representative *scospondin^icm15/icm15^* mutant and a control sibling at 20, 22, 24, 28 and 30 hpf in the revised version of the manuscript as Figure 1—figure supplement 1, which further confirms our previous observation regarding the onset of the curled down phenotype (Cantaut-Belarif et al., 2018).

4) Detailed protocols are needed.

We now detailed the protocols in the revised version of the manuscript, especially drug treatments, brain ventricle injections and immunohistochemistry procedures (see the Materials and methods section).

5) Editorial revision should include comments on similarities and differences with Lu et al.

We have now highlighted the similarities and differences with Lu et al. in the revised version of the manuscript.

Similarly to Lu et al., 2020: we found that in *scospondin* mutants epinephrine can compensate for the loss of the Reissner fiber on body axis curvature defects and *urp* neuropeptides expression. However, we add to this previous observation that norepinephrine can also act the same (see the Discussion subsection “Adrenergic activation restores the Reissner fiber-dependent signaling and axis straightening”).

Note that in contrast to Lu et al., we used here a quantitative approach to measure the effect of both monoamines on *urp* gene expression (using quantitative PCR on numerous independent samples instead of *in situ* hybridization) and body axis curvature (systematic ear-to-tail angle measurements on independent experimental groups).

In contrast to Lu et al., 2020: we report here a downregulation of *urp2* gene expression *only* and not *urp1* in *scospondin* mutants (subsection “*urp2* expression in ventral CSF-cNs depends on the presence of the Reissner fiber and impacts on the curvature of embryonic axis”). Thus, we contradict the idea that the Reissner fiber is necessary for the expression of both neuropeptides and rather show that *urp2* expression only is dependent on the Reissner fiber.